# Quasi-one-dimensional hydrogen bonding in nanoconfined ice

Pavan Ravindra[1,2,7], Xavier R. Advincula [1,3,4,7], Christoph Schran [3,4], Angelos Michaelides [1,4] ✉ & Venkat Kapil [1,4,5,6] ✉

The Bernal-Fowler ice rules stipulate that each water molecule in an ice crystal should form four hydrogen bonds. However, in extreme or constrained conditions, the arrangement of water molecules deviates from conventional ice rules, resulting in properties significantly different from bulk water. In this study, we employ machine learning-driven first-principles simulations to identify a new stabilization mechanism in nanoconfined ice phases. Instead of forming four hydrogen bonds, nanoconfined crystalline ice can form a quasi-one-dimensional hydrogen-bonded structure that exhibits only two hydrogen bonds per water molecule. These structures consist of strongly hydrogen-bonded linear chains of water molecules that zig-zag along one dimension, stabilized by van der Waals interactions that stack these chains along the other dimension. The unusual interplay of hydrogen bonding and van der Waals interactions in nanoconfined ice results in atypical proton behavior such as potential ferroelectric behavior, low dielectric response, and long-range proton dynamics.

Nanoconfined water is prevalent in biological and geological systems, and it has technological applications in tribology[1], desalination[2], and clean energy[3]. Experiments have revealed a diverse range of unusual properties in these systems containing confined water[4,5]. While a unified theory of the thermodynamics of systems confined to nanoscale dimensions is missing[6,7], atomistic simulations have aided our interpretation of experiments by revealing the microscopic structure and dynamics of nanoscale water[8–11]. Specifically, first-principles simulations provide a bottom-up format for predicting the physicochemical behavior of water in complex conditions[12–17] where empirical models fitted to bulk data may be insufficient.

Recent first-principles studies, direct[15] or machine-learning-accelerated[16,17], have explored finite-temperature phase behaviors of nanoconfined water in atomistically flat nanocavities – meant to resemble the conditions of water trapped between two graphene sheets. Notably, Kapil et al.[16] calculated the temperature-pressure phase diagram of a monolayer film of water by using machine learning potentials (MLPs)[18,19]. They unveiled crystalline monolayer phases comprising hexagonal, pentagonal, and rhombic motifs, an entropically stabilized hexatic phase of water, and a superionic phase at dramatically lower temperatures and pressures than its bulk counterpart[20–23]. More recently, Lin et al.[17] extended this investigation to larger confinement widths and a wider range of confinement pressures, revealing new bilayer ice phases. An intriguing aspect of these nanoconfined ice phases is their deviation from the Gibbs-Thomson relation[16,17]. These phases aren't simply bulk ice phases with altered phase boundaries; rather, they consist of unique motifs and topologies that are not observed in bulk ice.

Given the significant differences in the structure of bulk and nanoconfined water, it is essential to understand the underlying principles that distinguish their phase behaviors. The behaviors of crystalline ice are well explained using Pauling's principle of hydrogen

[1]Yusuf Hamied Department of Chemistry, University of Cambridge, Lensfield Road, Cambridge CB2 1EW, UK. [2]Department of Chemistry, Columbia University, 3000 Broadway, New York, NY 10027, USA. [3]Cavendish Laboratory, Department of Physics, University of Cambridge, Cambridge CB3 0HE, UK. [4]Lennard-Jones Centre, University of Cambridge, Trinity Ln, Cambridge CB2 1TN, UK. [5]Department of Physics and Astronomy, University College London, 17-19 Gordon St, London WC1H 0AH, UK. [6]Thomas Young Centre and London Centre for Nanotechnology, 19 Gordon St, London WC1H 0AH, UK. [7]These authors contributed equally: Pavan Ravindra, Xavier R. Advincula. ✉e-mail: am452@cam.ac.uk; v.kapil@ucl.ac.uk

bonding[24] and the Bernal-Fowler ice rules, which state that each water molecule should participate in four total hydrogen bonds[25]. However, these ice rules can be broken under extreme conditions and through translational symmetry disruptions at interfaces and in nanoscale confinement. These violations result in a variety of unusual phase behaviors, including a quantum delocalized state of water across nanocapillaries[26], a square-like ice phase encapsulated between graphene sheets[27], thin epitaxial films of ice exhibiting ferroelectric nature[28], and the formation of water chains[29] and nanoclusters resembling cyclic hydrocarbons on metals[30]. Attempts have been made to extend the Bernal-Fowler ice rules to these settings, and such modified ice rules generally focus on incorporating water-surface interactions into the hydrogen bonding picture of the Bernal-Fowler ice rules[12,30,31]. However, several recently predicted nanoconfined ice phases exhibit fewer than four hydrogen bonds per water molecule[15–17], which brings into question the extent to which nanoconfined water phases are stabilized by hydrogen bonding and if such modified ice rules should be centered around hydrogen bonding at all.

In this work, we utilize first-principles calculations and MLPs to identify a new stabilization mechanism in nanoconfined ice phases beyond conventional ice rules. Instead of forming four hydrogen bonds per water molecule, nanoconfined ice phases can form zigzagging quasi-one-dimensional hydrogen-bonded chains stacked together by van der Waals (vdW) interactions. We identify this mechanism across monolayer and bilayer crystalline ice found in refs. [15–17] in confinement slits of varying sizes. Employing MLPs to access long timescales at significantly lower computational costs than traditional first-principles methods, we show that quasi-one-dimensional hydrogen bonding is stable at finite temperatures up to the melting point of nanoconfined ice. The arrangement of strong and weak interactions in nanoconfined water bears similarities to two-dimensional vdW materials consisting of so-called quasi-one-dimensional chains[32,33]. Conventionally, these materials comprise strong covalent bonds within the quasi-one-dimensional chains (as opposed to hydrogen bonds), with weak inter-chain vdW attraction holding these chains together. These materials exhibit unusual electronic, vibrational, and optical properties[32,33], potentially exhibiting superconductivity at higher pressures[34]. Analogously, the intrinsic anisotropic bonding in the flat-rhombic phase leads to anomalous rotational dynamics of water molecules, including detectable long-range concerted disorder on the scale of nanometers, which has potential applications for molecular devices.

## Results

### Violations of conventional ice rules lead to a quasi-one-dimensional hydrogen bonded ice phase

In bulk ice, the Bernal-Fowler ice rules dictate that each water molecule in a stable ice phase should participate in four hydrogen bonds with its neighboring water molecules: 2 donated and 2 accepted hydrogen bonds[25]. Previous work on nanoconfined water at the DFT level has suggested that the thermodynamically stable monolayer phase is a square structure that still exhibits this 4-fold coordination[9,12]. However, more recent first-principles-level simulations have suggested a wide variety of stable phases of nanoconfined water spanning a broad range of different temperature and lateral pressure conditions[14–17]. Kapil et al.[16] report a hexagonal phase stable below 0.1 GPa, a pentagonal phase stable from 0.1 to 0.5 GPa, and a flat-rhombic phase stable above 0.5 GPa for a confinement width of 5 Å. Reference[16] also confirms the greater thermodynamic stabilities of these phases with respect to the square phase using the chemically accurate and robust Quantum Monte Carlo (QMC) calculation setup[35,36]. References[15,17] explored even higher lateral pressures for a confinement width of 6 Å. These studies found that the stable phase beyond 0.5 GPa is a zigzag monolayer ice (ZZMI) phase, which is topologically equivalent to the flat-rhombic phase, except that the larger confinement width leads to a small buckling of the oxygen atoms. In the 15–20 GPa regime, Lin et al. report a zigzag quasi bilayer ice (ZZ-qBI) structure as the stable phase[17]. This phase also resembles the flat-rhombic phase, although the buckling in this phase results in two nearly distinct layers of water molecules.

As shown in Fig. 1a and Table 1, none of the above thermodynamically stable nanoconfined ice phases satisfy conventional ice rules: all of these phases display fewer than four hydrogen bonds per water molecule. However, the water molecules in each of these phases violate the ice rules in different ways. In the low-pressure hexagonal phase, all water molecules exhibit a 3-fold hydrogen bonding coordination: half of the water molecules donate two and accept one hydrogen bond, and the other half donate one and accept two. Water molecules in the intermediate-pressure pentagonal phase exhibit diverse hydrogen bonding motifs, with coordination numbers ranging

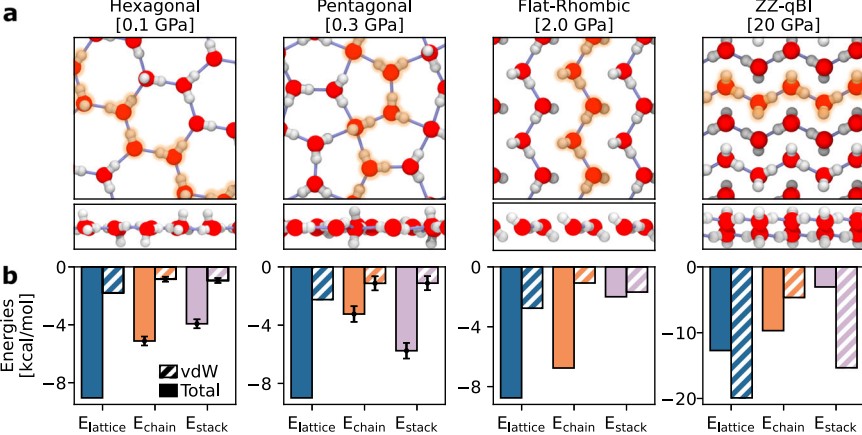

**Fig. 1 | Nanoconfined ice phases and the role of vdW stacking of hydrogen bonded chains. a** The four nanoconfined ice phases considered in this work from refs. 16,17, along with the lateral pressures used for simulating each phase. Solid lines between water molecules indicate hydrogen bonds. We highlight chains of hydrogen-bonded water molecules in each phase. For estimating $E_{chain}$, we use the unique highlighted chains in the case of the flat-rhombic and ZZ-qBI phases. For the hexagonal and pentagonal phases, we report the average value and the standard deviation as the error bar, where the standard deviation is computed across different choices of chains, as described in the text. **b** For each phase, we compute the stabilization energy of a single molecule in the 0 K crystal structure, $E_{lattice}$; the (average) stabilization energy of a water molecule within the hydrogen bonded chain(s), $E_{chain}$; and the stabilization energy between chains in the crystal structure, $E_{stack}$. The hatched bars show the contribution of vdW interactions to the corresponding stabilization energy. Source data are provided as a Source Data file.

**Table 1 | First-principles calculations for four nanoconfined ice phases**

| Phase | Geometric hydrogen bonds | | | $E_{lattice}$ (kcal/mol) | | $E_{chain}$ (kcal/mol) | | $E_{stack}$ (kcal/mol) | |
|---|---|---|---|---|---|---|---|---|---|
| | Donors | Acceptors | Total | Total | vdW | Total | vdW | Total | vdW |
| Hexagonal | 1.5 | 1.5 | 3.0 | −9.05 | −1.79 | −5.12 ± 0.31 | −0.85 ± 0.17 | −3.93 ± 0.31 | −0.94 ± 0.17 |
| Pentagonal | 1.6 | 1.6 | 3.2 | −8.99 | −2.24 | −3.20 ± 0.50 | −1.10 ± 0.50 | −5.80 ± 0.50 | −1.10 ± 0.50 |
| Flat-rhombic | 1.0 | 1.0 | 2.0 | −8.75 | −2.76 | −6.76 | −1.08 | −2.00 | −1.68 |
| ZZ-qBI | 1.0 | 1.0 | 2.0 | −12.71 | −19.91 | −9.69 | −4.65 | −3.05 | −15.33 |

Average number of hydrogen bonds and energies of the hexagonal phase at 0.1 GPa, the pentagonal phase at 0.3 GPa, the flat-rhombic phase at 2.0 GPa, and the ZZ-qBI phase at 20 GPa in their equilibrium structures. We report the average number of donated, accepted, and total hydrogen bonds per water molecule. The energy values include the total and vdW contributions to the lattice energy $E_{lattice}$, the cohesive energy of a quasi-one-dimensional chain of water molecules $E_{chain}$, and the remaining stabilization energy from interactions between chains of water molecules $E_{stack}$. The equilibrium structures of the ice phases were obtained from fixed-cell geometry optimizations, as explained in the methods section. The energies were obtained directly from first-principles calculations. Hydrogen bonds were detected using the standard geometric criteria from ref. 68. For the hexagonal and pentagonal phases, we report the average value and the standard deviation as the error bar, where the standard deviation is computed across different choices of chains, as described in the text.

from 2 to 4. This results in an average hydrogen bond coordination of 3.2 in the pentagonal crystal structure. The high-pressure flat-rhombic (or the ZZMI) phase exhibits a single hydrogen-bonding motif in which each water molecule donates one and accepts one hydrogen bond. The resulting hydrogen bonding network contains just two hydrogen bonds per water molecule, deviating significantly from the conventional ice rules. The ZZ-qBI phase exhibits a similar network, with just two hydrogen bonds per water molecule. Since this phase is stable even at lateral pressures much higher than the flat-rhombic phase, this suggests that such extreme violations of the bulk ice rules are likely to occur in high lateral pressure conditions.

The flat-rhombic phase is thermodynamically stable in a sizeable temperature-pressure region of the phase diagram[15–17] despite having just two hydrogen bonds per molecule. Considering that ice in general is primarily stabilized by a dense network of hydrogen bonds[24,37], this stability over a broad range of conditions is highly unusual. To understand this unexpected behavior, we examine the structure of the flat-rhombic phase and look for similarities with other types of materials. As depicted in Fig. 1a, the flat-rhombic phase is characterized by a zigzagging hydrogen bond network that runs along quasi-one-dimensional chains[15,17]. These chains are stacked alongside each other without any hydrogen bonding between chains. This structure is similar to quasi-one-dimensional vdW materials[32,33], which exhibit unusual electronic, optical, and vibrational properties. However, while these materials consist of covalently bonded chains held together by weak inter-chain vdW interactions, the chains in the flat-rhombic phase are stabilized by hydrogen bonding.

To determine whether the flat-rhombic phase can be classified as a quasi-one-dimensional vdW material, we calculate the binding energy of the one-dimensional chains and study the role of vdW interactions in stabilizing these chains. In Fig. 1b and Table 1, we report three different types of binding energies of a water molecule in each phase: the stabilization energy of a water molecule in the corresponding crystal structure, $E_{lattice}$; the stabilization energy of a water molecule within a continuous chain of hydrogen-bonded water molecules, $E_{chain}$ (see highlighted chains of water molecules in Fig. 1a); and the remaining stabilization energy from interactions between chains of water molecules in each phase, $E_{stack}$. This means that the stacking energy is the stabilization energy of the lattice that is not captured by the chains alone: $E_{stack} = E_{lattice} - E_{chain}$. In the hexagonal phase, we consider two distinct chain types in the 'zigzag' and 'armchair' directions and report the average binding energies. As can be seen in the pentagonal phase snapshot in Fig. 1a, there is no clear choice for a unique hydrogen bond chain in the pentagonal phase. This shows qualitatively that the pentagonal phase cannot clearly be separated into distinct hydrogen bond chains. Nonetheless, to perform the quantitative comparison shown in Fig. 1b, we averaged our energies over three different hydrogen bond chains in the pentagonal phase. The energies for these different chains were all extremely close to each other, suggesting that this quantitative comparison is robust with respect to the exact choice of hydrogen bond

chain in the pentagonal phase. For the flat-rhombic and ZZ-qBI phases, we considered a single unique chain.

In Fig. 1b, we observe that $E_{chain}$ contributes to approximately 50% or less of the crystal stabilization energy in both hexagonal and pentagonal phases. Therefore, these phases have high stacking energies between chains, stabilized primarily by hydrogen bonds, with vdW interactions (indicated by hatched regions) playing only a secondary role in stabilizing the chains. Conversely, in the flat-rhombic phase, the primary crystal stabilization source is from the chains. Here, the stacking energy is low, with roughly 90% of the stacking energy resulting from vdW interactions. This difference indicates that the interactions stabilizing water molecules in the flat-rhombic phase contrast starkly with those in the hexagonal and pentagonal phases. The flat-rhombic phase appears to form hydrogen bonds in one dimension while being stabilized by weak vdW forces in the other, classifying it as a quasi-one-dimensional vdW crystal. Similarly, the stacking energy of the ZZ-qBI phase is almost entirely driven by vdW interactions, meaning that vdW forces also stabilize the interactions between this phase's hydrogen-bonded chains. In fact, in the absence of vdW stacking, the lattice energy is positive, meaning that the ZZ-qBI crystal structure would be thermodynamically unstable. This suggests that these vdW interactions between hydrogen-bonded chains play an essential role in stabilizing nanoconfined crystal structures at high lateral pressures. As our MLP isn't trained for pressures beyond 8 GPa, we are unable to comment on the dynamical stability or metastability of the ZZ-qBI phase. Similarly, in order to comment on the true dynamical stability or metastability of any of these phases in the absence of vdW interactions, we would need to retrain an MLP at the revPBE0 level (in the absence of the D3 dispersion correction).

## Anomalous finite-temperature dependence of hydrogen bonding

The characteristic interactions in the flat-rhombic phase lead to anomalous finite-temperature hydrogen bonding structure and dynamics. In bulk or conventionally hydrogen-bonded ice phases, such as the hexagonal and pentagonal monolayer phases, the most probable molecular orientations are the ones that maximize the number of hydrogen bonds[25], and the protons maximize their residence time in orientations with the maximum possible hydrogen bonds. However, in the flat-rhombic phase, even at finite temperatures (extending past 300 K), the most probable molecular orientations are not those that maximize hydrogen bonds but instead, those that maintain a distinct network of quasi-one-dimensional chains.

To characterize the rotational motion of individual water molecules, we define $\phi$ and $\theta$ angles for molecular dipoles. These are depicted in Fig. 2a and b, respectively. These figures also indicate the coordinate axes used to define these angles. The $\phi = \frac{\pi}{2} - \arccos(\hat{\boldsymbol{\mu}} \cdot \hat{\mathbf{z}})$ angle captures the alignment between the molecular dipole vector $\boldsymbol{\mu}$ and the z-axis, with hats above vectors indicating normalized unit vectors. A water molecule with $\phi = \pm \pi/2$ will have its dipole vector

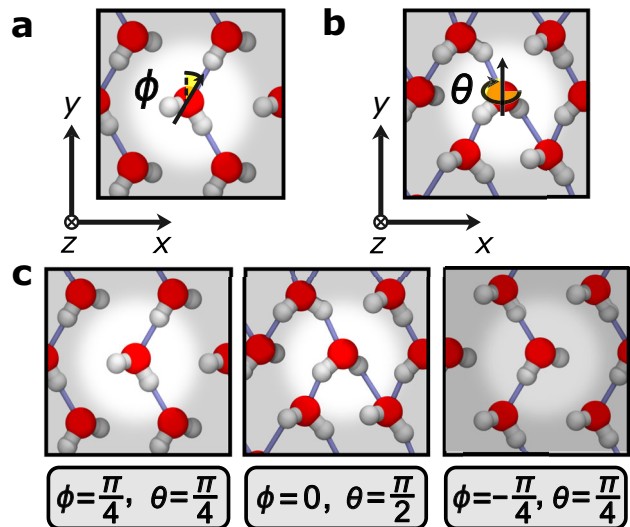

**Fig. 2 | Depiction of the orientations of water molecules. a** The $\phi$ angle captures in-plane and out-of-plane fluctuations of water molecules' dipole moments. **b** The $\theta$ angle captures in-plane rotations of water molecules. The mathematical definitions of the $\phi$ and $\theta$ angles are provided in the text. **c** The left and right images depict the molecular orientations at the probability maxima in Fig. 3a, while the middle image depicts the molecular orientation at the maximum for the number of putative hydrogen bonds in Fig. 3b.

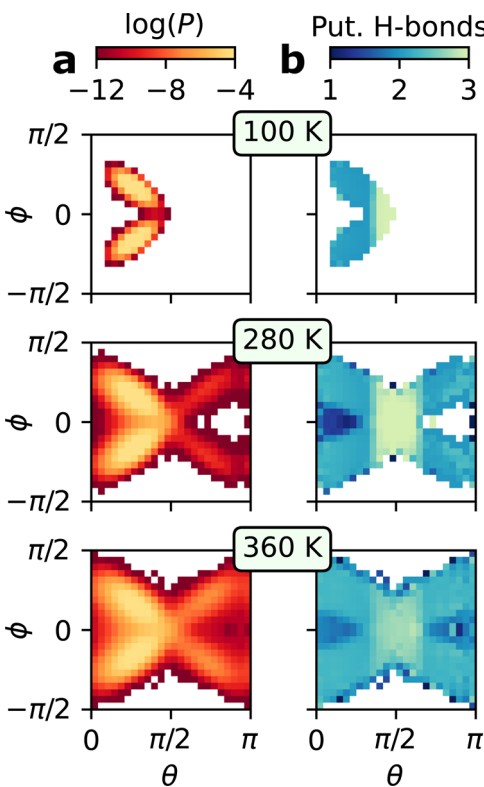

**Fig. 3 | Molecular orientation and the number of putative hydrogen bonds. a** The two-dimensional log probabilities along the $\phi$ and $\theta$ parameters for the flat-rhombic phase at 2 GPa and three different temperatures. **b** The average number of putative hydrogen bonds associated with each ($\theta$, $\phi$) molecular orientation for the flat-rhombic phase under the same conditions. Relevant molecular orientations are depicted in Fig. 2c. We employ the geometric definition of the hydrogen bond from ref. 68. Source data are provided as a Source Data file.

aligned with the $\pm z$-axis. We also wrap around the $\phi$ angles computed such that $\phi$ always lies in the range $[-\frac{\pi}{2}, +\frac{\pi}{2}]$. The $\theta = \arctan(\mathbf{c} \cdot \hat{\mathbf{z}} / \mathbf{c} \cdot \hat{\mathbf{x}})$ angle captures a molecule's rotation about its center of mass around the $y$ axis with $\mathbf{c} = \mathbf{r}_{OH_1} \times \mathbf{r}_{OH_2}$ and $\mathbf{r}_{OH_1}$ and $\mathbf{r}_{OH_2}$ being the OH bond vectors of an individual water molecule. $\theta$ measures the rotation of $\mathbf{c}$ around the $y$-axis. Since the choice of $H_1$ and $H_2$ is arbitrary, we wrap around the $\theta$ angles so that they always lie in the range $[0, \pi]$.

To illustrate the flat-rhombic phases' apparent hesitancy to form additional hydrogen bonds, we computed the probabilities of molecular orientations defined by these $\theta$ and $\phi$ angles. These two-dimensional log probabilities are juxtaposed with the mean number of hydrogen bonds for each two-dimensional bin determined by the $\theta$ and $\phi$ angles, as depicted in Fig. 3. As shown in Supplementary Note V, the nanoconfined hexagonal and pentagonal phases exhibit the expected trend: molecular orientations with more hydrogen bonds should be more stable. In contrast, the most probable flat-rhombic molecular orientations (depicted in the left and right plots of Fig. 2c) correspond to only two hydrogen bonds. This is true even though orientations with three hydrogen bonds (like the one in the middle plot of Fig. 2c) are possible. We begin to observe these orientations with three hydrogen bonds via the entropic exploration of $\theta$ and $\phi$ angles at high temperatures and in the presence of quantum nuclear fluctuations (see Supplementary Note VI). However, these orientations still do not emerge as local probability maxima, suggesting they are not stable or long-lived states. This behavior reinforces the observation that the two-dimensional flat-rhombic ice phase prefers to remain hydrogen bonded only along one dimension, even at finite temperatures.

The peculiar hydrogen bonding behavior of the flat-rhombic phase hence emphasizes the importance of incorporating dynamical information into the hydrogen bonding definition, as is needed in, e.g., the description of supercritical water[38]. As shown in Fig. 4a, if we employ a purely geometric definition of a hydrogen bond, counting the number of instantaneous *putative* hydrogen bonds that exist on average, we observe an apparent increase in the number of hydrogen bonds with temperature. In the same figure, we also observe that nuclear quantum effects apparently exhibit an additional 0.2 putative hydrogen bonds, even at temperatures as low as 20 K. This is the result

of the zero-point fluctuations lowering the energy barrier for sampling these additional hydrogen-bonded configurations, as we elaborate upon in Supplementary Note VI. In these configurations, hydrogen bonds form between different chains in the flat-rhombic crystal structure, as they do classically at high temperatures. The increased disorder due to quantum nuclear effects is not surprising, as the effect of quantum nuclear motion on properties of water has often been mapped to a temperature increase[39]. To characterize the hydrogen bonding motifs in this structure, we introduce the notation $N$D$M$A to indicate a water molecule that donates $N$ hydrogen bonds and accepts $M$ hydrogen bonds. Using this notation, we see that this increase in the number of putative hydrogen bonds can be attributed to the increasing prevalence of instantaneous 2D1A motifs, i.e. instantaneous hydrogen bonds between chains.

This apparent increase emerges from the shortcoming of the standard geometric definition of a hydrogen bond, which cannot distinguish fluctuating molecular orientations from long-lived orientations that correspond to true hydrogen bonds. As shown in Fig. 4c, the lifetime of the 2D1A orientations is extremely short – on the order of femtoseconds. As suggested by Schienbein and Marx[38], these fleeting hydrogen bonds are best referred to as either putative or geometric, as they do not last long enough to be considered hydrogen bonds from a traditional perspective. In Supplementary Note II, we compute the number of hydrogen bonds that survive for typical intermolecular oscillations. Within this definition by Schienbein and Marx[38], a hydrogen bond is not counted if its lifetime is shorter than the period of typical intermolecular oscillations. The resulting dynamical hydrogen bond count indeed exhibits the expected disorder-induced decrease as we raise temperature.

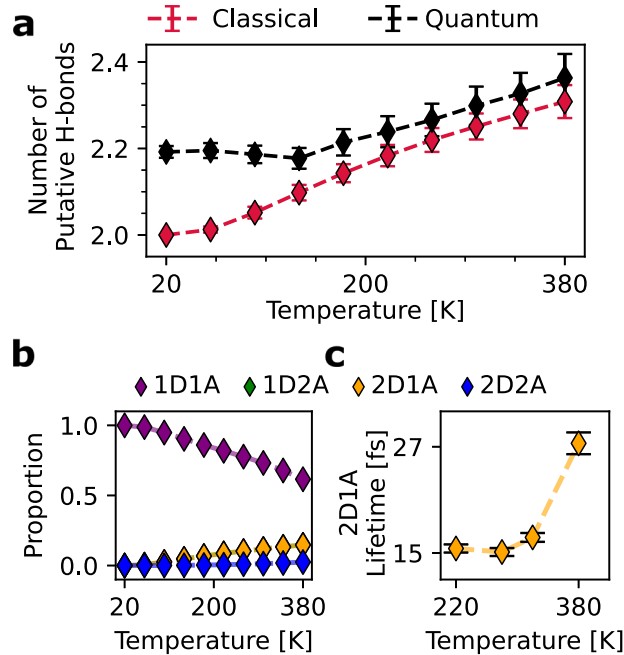

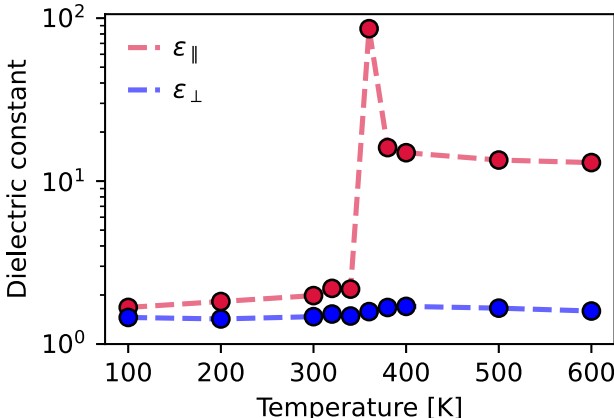

**Fig. 4 | Temperature dependence of hydrogen bonding. a** The average number of putative/geometric hydrogen bonds across the range of temperatures in which the flat-rhombic phase is stable using classical and quantum simulations. The error bars show the standard error of the mean, as computed by block averaging over 10 blocks. **b** The proportion of water molecules of each hydrogen bonding motif (defined in the text). The 1D2A proportions are perfectly aligned with the 2D1A proportions. **c** The 2D1A motif's lifetime remains extremely short across the range of temperatures in which it is observed. The error bars show the standard error of the mean, computed across all instances of hydrogen bonding. Dashed lines serve as a visual guide for the eye. Source data are provided as a Source Data file.

**Fig. 5 | Temperature dependence of the dielectric response.** The classical dielectric constant of the flat-rhombic phase across the range of temperatures considered in this work. The singularity at 380 K corresponds to the transition to the hexatic phase of nanoconfined water[53]. Source data are provided as a Source Data file.

## Dielectric nature of the flat-rhombic phase

A consequence of the quasi-one-dimensional hydrogen-bonded structure is that it may possess ferroelectric behavior due to its net dipole moment along the direction of the hydrogen-bonded chains. Ferroelectricity in ice has been conjectured previously in force field simulations[40], first-principles calculations of nanoconfined water[41], and experiments on confined[28] and supported[42] films. To investigate the dielectric behavior of flat-rhombic ice, we model the temperature dependence of its dielectric response in Fig. 5. We analyzed the in-plane $\varepsilon_\parallel$ and out-of-plane $\varepsilon_\perp$ dielectric constants based on the variance of the system's polarization. Since our MLP only predicts the potential energy surfaces, a first principles investigation of the dielectric response would be computationally demanding due to numerous single point calculations of the electronic polarization[43]. Therefore, for a semi-quantitative understanding, we use a simple linear polarization model based on TIP4P charges[44].

We select the TIP4P water model due to its low computational cost and a semi-quantitative description of the dielectric response of bulk[44] and confined[45] water. The TIP4P model underpredicts the polarization of aqueous systems as it doesn't incorporate the electronic polarization of the water molecules. For instance, TIP4P predicts a molecular dipole moment of 2.348 D which is 12.8% lower compared to first principles estimates[46], and effectively leads to a ~ 25% lower dielectric response. To incorporate the lack of the electronic polarization in the TIP4P model, we rescale the calculated dielectric response by a factor of 1.25, as has been done previously[45]. We report the classical dielectric response, as quantum nuclear effects are expected to only make a small quantitative difference[44].

We observe low and near-constant in-plane and out-of-plane dielectric constants for the flat-rhombic phase, in agreement with the

experiments on nanoconfined water from ref. 47. The low dielectric response of this phase is only an indirect outcome of vdW interactions, as they dictate the thermodynamic stability of the flat rhombic phase. The dielectric nature of flat-rhombic ice remains the same from 0 K up to its phase transition into the paraelectric hexatic (disordered) water phase[16]. The temperature independence of the flat-rhombic phase's dielectric behaviors reflects the resilience of the quasi-one-dimensional structure to thermal and quantum nuclear fluctuations. The singularity observed in Fig. 5 indicates the temperature at which the phase transition to the hexatic phase occurs. The hexatic phase exhibits an increased in-plane dielectric constant but a similarly small out-of-plane dielectric constant.

## Long-ranged spatial ordering and coherent proton dynamics

Thus far, the results in this work have considered large simulation boxes with 576 molecules spanning tens of nanometers. However, in simulation boxes containing 144 molecules in a relatively small but experimentally accessible system size on the order of nanometers[48,49], we identify unusual concerted dynamics at intermediate temperatures involving the breaking and forming of hydrogen bonds between equivalent structures. In this concerted motion, we observe an exchange between the individual molecular $\phi = \pm \pi/4$ states as a collective motion throughout the entire system, in which all the water molecules simultaneously swap the signs of their $\phi$ angles (see Supplementary Movie 1).

To describe this concerted motion, we define a $\sigma$ order parameter (see Supplementary Note III) to distinguish the two states shown in Fig. 6a. The structural change to the flat-rhombic phase between $\sigma$ values can be seen in Fig. 6a. When the sign of $\sigma$ changes, the directions of the dipoles in each row switch in a concerted fashion. During this motion, we also observe the complete rearrangement of the hydrogen bonding network in the lattice. Figure 6a shows labeled water molecules in each $\sigma$ state, illustrating that the hydrogen bonds in these two states are between different pairs of water molecules. At low temperatures, the system remains frozen in a single $\sigma$ value. The resulting $\sigma$ free energy profile contains two minima separated by a large free energy barrier, shown in Fig. 6c. At intermediate temperatures, we begin to see exchanges between $\sigma$ signs on the order of ~10 ps, as shown in Fig. 6b. Compared to the low-temperature setting, the intermediate temperature free energy profile exhibits a clear finite free energy barrier between these states. Eventually, at high enough temperatures, the free energy profile along $\sigma$ exhibits an extremely small free energy barrier, allowing the system to explore molecular configurations freely. While the highest temperatures explored here are

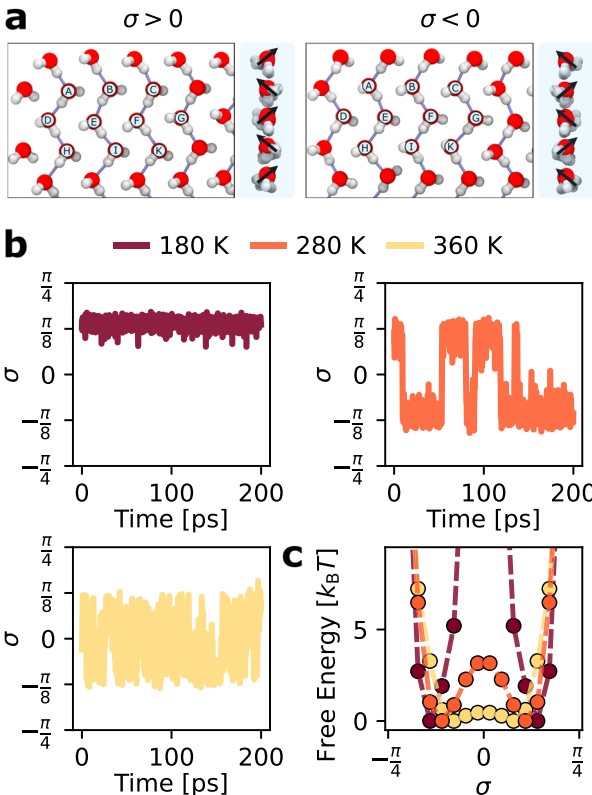

**Fig. 6 | Hydrogen bonded switching behavior. a** The two symmetry-related structures of the flat-rhombic phase, with alternating rows of aligned dipoles, discerned by the different signs of $\sigma$. A zero value of $\sigma$ indicates that the molecular dipoles are not aligned row-wise. The side views in each panel show the system as viewed from the right side, where the switch in the row-dependent dipole direction between the two states is clear. **b** For smaller unit cells with 144 water molecules, we observe coherent switching between these two states at intermediate temperatures, as shown in the $\sigma$ trajectories for three different temperatures. **c** The free energy profiles for this $\sigma$ parameter show that raising the temperature lowers the free energy barrier between these states. The smooth, low-barrier free energy profile at high temperatures indicates that the molecular dipoles lose spatial coherence. Source data are provided as a Source Data file.

greater than the melting temperature of ice, we believe that our trajectories correspond to a metastable state associated with the solid phase. Since the free energy profiles at the intermediate and high-temperature conditions are qualitatively the same, the near-free exploration of molecular configurations in the high-temperature simulations is a result of thermal activation. To confirm this hypothesis, we compute rotational autocorrelation functions at these different temperatures in Supplementary Note IV.

We also note that quantum nuclear motion makes the system more disordered due to zero-point fluctuations lowering the barrier for the system's hydrogen bond network to switch between the free energy minima, as shown in Supplementary Fig. 8. Furthermore, since nuclear quantum effects lead to increased proton disorder, the $\sigma$ parameter gets pushed closer to 0 on average. This causes the minima in the free energy profiles along $\sigma$ to move closer to 0 in our PIMD simulations as compared to the free energy profiles from our classical simulations.

As stated earlier, this concerted motion is not observed in the larger simulation cell size that contains 576 molecules for the timescales we consider: it is only seen in simulations with 144 water molecules. This suggests that the free energy barrier for this concerted motion is size-extensive. As a result, we would expect a larger system size to either exhibit an ordered phase or a phase in which smaller

domains that exhibit this behavior coexist. However, crystallites of nanoconfined water molecules containing around 100 water molecules are experimentally accessible, so such concerted motion could be observed in experiments as small ice crystallites forming between graphene sheets[48] or high-pressure graphene nanobubbles formed by irradiation[49]. Previous work has suggested that the simulation setup we employ is still reasonable for studying such encapsulated systems, even without explicit confining atoms[48,50]. A careful treatment of such encapsulated systems would require a detailed analysis of the water-carbon interactions at the edge of the confined water pockets. In this work, we ignore these edge effects and instead focus on characterizing the behavior of nanoconfined water under idealized atomistically flat nanoconfinement. A careful consideration of the edge effect would be an interesting and relevant topic for future work.

## Discussion
We have analyzed the hydrogen bond structure and dynamics in the flat-rhombic phase of monolayer nanoconfined ice at first-principles-level accuracy using machine learning-driven simulations. Our work not only corroborates previous experimental and theoretical studies showing violations of ice rules at interfaces[15,17,26,28,51] but also highlights an extreme scenario where hydrogen bonding plays a limited role in stabilizing the crystalline structure of ice. General ice rules must account carefully for the enthalpic penalty of breaking a hydrogen bond due to interactions with the confining walls and favorable vdW interactions. Instead of maximizing the number of hydrogen bonds, flat-rhombic ice forms one-dimensional chains stabilized by vdW interactions akin to quasi-one-dimensional vdW functional materials[32,33]. The ZZ-qBI phase identified in ref. 17 is stable at higher lateral pressures than the flat-rhombic phase, yet it exhibits a similar behavior, indicating that such hydrogen bond topologies may occur across a broad range of nanoconfined conditions. This quasi-one-dimensional ice structure is robust with respect to thermal fluctuations and the rotational dynamics of water molecules, forming two or fewer hydrogen bonds up to its transition to the hexatic phase. We anticipate that direct evidence of the unique structure of the flat-rhombic phase could be experimentally investigated using sum-frequency generation spectroscopy[52] by disentangling the O–H vibrational stretching band[53] into donor-acceptor contributions.

Like quasi-one-dimensional vdW functional materials that exhibit unique electronic, vibrational, and optical properties[32–34], the flat-rhombic phase exhibits unusual properties with technological prospects. We observe long-ranged ordering of molecular dipoles along with a high spatial coherence in the hydrogen bond dynamics in the flat-rhombic ice phase, which is typically uncommon in systems containing on the order of 100s of molecules. This behavior is consistent with the characteristics of a two-dimensional Ising model system. We anticipate that this concerted motion in flat-rhombic ice patches could be used to guide directional behaviors in nanoscale molecular devices that have already shown success in performing macroscopic-level tasks at surfaces by exploiting directed translational motion on the nanometer scale[54]. Coherent proton dynamics observed experimentally in bulk systems, such as proton tunneling in hexagonal ice[55] and dielectric phase transitions in molecular ferroelectrics[56,57], have already been computationally detected on much smaller scales[58–60]. This suggests that confinement effects might be crucial in phenomena like tunneling-induced or dielectric phase transitions. Our work opens doors to exploring new confined molecular materials with the structure-function relationships of low-dimensional functional materials that exhibit technologically relevant properties.

## Methods
### Confining potential
The nanoconfined system considered in this work is a single layer of water molecules trapped between two parallel sheets – mimicking the

experimental setup in ref. 27. We model the interactions between the sheets and the water molecules using a simple Morse potential fit to water-carbon QMC interaction energies[61]. Such a potential, characterized by perfectly smooth walls, has been widely adopted in previous studies, both in force field[9–11] and first-principles research[12,14–17,62]. The uniform confinement potential has demonstrated semi-quantitative accuracy in describing the behavior of water confined within graphene-like cavities[11], as corroborated by a good agreement of stable phases and melting temperatures with respect to confinement simulations that include explicit carbon atoms[50]. Furthermore, the uniform confinement model's atomistically flat nature allows for a clean interpretation of topological confinement effects – a phenomenon that extends beyond the specific context of graphene-based confinement.

### Machine learning potential

For water-water interactions, we employ a newly-trained MLP[18,19] at the revPBE0-D3 level[63,64] – an appropriate dispersion-corrected hybrid-functional DFT level for bulk, interfacial and nanoconfined water[16,53,65,66]. We report the training and validation protocols for the MLP in Supplementary Note IB. Our simulations are run in the $NP_{xy}T$ ensemble, where $P_{xy}$ denotes the lateral pressure, or vdW pressure[48], that acts in the $x$ and $y$ directions. This lateral pressure emulates the net lateral forces on the water molecules due to the edges of the confining pocket (further details on the origins of this lateral pressure can be found in Supplementary Note IC). Both simulations[50] and experiments[48] have estimated this vdW pressure to be on the gigapascal scale. Hence, our simulations are run at a lateral pressure of 2.0 GPa across various temperatures within the metastability range of the flat-rhombic phase[16]. We refer the reader to Supplementary Note I for more details on the simulations.

### First-principles calculations

To estimate the (zero temperature) static stabilization energy of a nanoconfined ice crystal or a quasi-one-dimensional chain of water molecules, we perform direct first-principles calculations using the CP2K code[67] employing the revPBE0-D3 functional and the convergence parameters from ref. 16. We perform these calculations on geometry-optimized structures of the confined ice phases[17]. The geometry-optimized structures of the hexagonal, pentagonal, and flat-rhombic monolayer phases were taken from ref. 16. In contrast, the structure of the zigzag quasi bilayer ice from ref. 17 was optimized using the CP2K code[67]. We define the stabilization energy of a crystal with respect to an isolated molecule as $E_{\text{lattice}} = E_{\text{crystal}} - E_{\text{gas}}$, where $E_{\text{crystal}}$ is the single-point energy of the crystalline lattice per water molecule, and $E_{\text{gas}}$ is the single-point energy of an isolated water molecule in vacuum. Similarly, we define the stabilization energy of a chain of water molecules with respect to an isolated molecule as $E_{\text{chain}} = E_{\text{chain}}^{\text{struct.}} - E_{\text{gas}}$, where $E_{\text{chain}}^{\text{struct.}}$ is the single-point energy of the chain structure, and the stabilization energy of a crystal with respect to a chain of water molecules as $E_{\text{stack}} = E_{\text{lattice}} - E_{\text{chain}}$. We also estimate these quantities without the D3 dispersion correction to assess the role of vdW interactions.

## Data availability

The data supporting the findings of this study are openly available on GitHub (https://github.com/water-ice-group/hbond_nanoconfined_water). This includes all simulation scripts and starting structures. They are also available via this GitHub repository. Source data are provided with this paper.

## Code availability

All of the analysis scripts supporting the findings of this study are openly available on GitHub (https://github.com/water-ice-group/hbond_nanoconfined_water).

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

## Acknowledgements

We thank Jinggang Lan, Dominik Marx, Philipp Schienbein, Michele Ceriotti, and all of A.M.'s research group members for their comments on the manuscript. P.R. would also like to thank David R. Reichman for his academic support during this project. V.K. acknowledges support from the Ernest Oppenheimer Early Career Fellowship and the Sydney Harvey Junior Research Fellowship, Churchill College, University of Cambridge. A.M. and X.R.A. acknowledge support from the European Union under the "n-AQUA" European Research Council project (Grant no. 101071937). C.S. acknowledges partial financial support from the Deutsche Forschungsgemeinschaft (DFG, German Research Foundation) project number 500244608. P.R. would like to thank The Winston Churchill Foundation of the United States for their financial support. P.R. would also like to acknowledge that: "This material is based upon work supported by the U.S. Department of Energy, Office of Science, Office of Advanced Scientific Computing Research, Department of Energy Computational Science Graduate Fellowship under Award Number DE-SC0024386. This report was prepared as an account of work sponsored by an agency of the United States Government. Neither the United States Government nor any agency thereof, nor any of their employees, makes any warranty, express or implied, or assumes any legal liability or responsibility for the accuracy, completeness, or usefulness of any information, apparatus, product, or process disclosed, or represents that its use would not infringe privately owned rights. Reference herein to any specific commercial product, process, or service by trade name, trademark, manufacturer, or otherwise does not necessarily constitute or imply its endorsement, recommendation, or favoring by the United States Government or any agency thereof. The views and opinions of authors expressed herein do not necessarily state or reflect those of the United States Government or any agency thereof." We are grateful for computational support from the Swiss National Supercomputing Centre under project s1209, the UK national high-performance computing service, ARCHER2, for which access was obtained via the UKCP consortium and the EPSRC grant ref EP/P022561/1, and the Cambridge Service for Data Driven Discovery (CSD3).

## Author contributions

P.R., A.M., and V.K. conceived the study. P.R., V.K., and C.S. trained the machine learning potential used in this work. P.R. and X.R.A. performed molecular dynamics simulations using this MLP. All of the authors were involved in discussing results and assembling the manuscript.

## Competing interests

The authors declare no competing interests.
