## [Peer Review File · Nature Communications]

Quasi-one-dimensional hydrogen bonding in nanoconfined iceReviewer #1 (Remarks to the Author):

Please see the attached file.

Reviewer #1 Attachment on the following page

Publish after major revision.

Comments:

In this manuscript, the author introduces a novel "flat-rhombic" ice characterized by one-dimensional water chains, with van der Waals interactions dominating the inter-chain interactions. Notably, each water molecule in this quasi-one-dimensional hydrogen-bonded monolayer ice forms only two hydrogen bonds, and the prevalence of van der Waals interactions leads to a deviation from the conventional ice rule. Additionally, the author reports a concerted motion behavior within the hydrogen bonding network of the "flat-rhombic" ice. These findings hold fundamental significance in water science and have widespread interest among researchers. However, certain critical issues need to be addressed before considering this manuscript for publication in *Nat. Commun.*

- 1) A more comprehensive discussion is needed to clarify the distinctions between the quasi-one-dimensional monolayer ice and the flat-rhombic phase presented by Kapil et al in Ref. 17, as well as the zigzag quasi-bilayer ice reported in Ref. 18. Particularly, it seems that the latter exhibits some critical similarities with the "flat-rhombic" ice reported here. For instance, both configurations consist of one-dimensional water chains connected through H-bonding, subsequently stacking together through van der Waals (vDW) forces (it is not mentioned in Ref. 18). Additionally, both ices involve water molecules forming only two hydrogen bonds. Is the "flat-rhombic" ice considered a special case existing exclusively within the 5 Å nanoslit?
- 2) The author claimed that the van der Waals interactions play a critical role in stabilizing the "flat-rhombic" ice. The definitions of stabilization energies lack clarity. Should $E_{lattice} = E_{crystal} - E_{gas}$ be understood as $E_{lattice} = E_{crystal} - N * E_{gas}$, where N is number of water molecules in a unitcell? Is $E_{crystal}$ the energy of the fully relaxed unit cell? Typically, the term "single-point energy" refers to the energy of the structure as it is built without any relaxation performed. Furthermore, to enhance the robustness of the conclusion, it would be beneficial to present an additional example where van der Waals interactions serve as the primary stacking force. The zigzag quasi-bilayer ice mentioned in Q#1 could serve as a potential candidate for such an illustration.
- 3) The author has described an interesting, concerted motion behavior within the hydrogen bonding network of the "flat-rhombic" ice in the system consisting of 144 molecules. Is this behavior dependent on the system size, and will it dissipate in larger systems?
- 4) Regarding the concerted motion mentioned in Q#3, the author computed the free-energy variation with the σ parameter. It would be beneficial to include additional details on the methodology employed to calculate the free-energy profiles depicted in Fig. 4c. The author also took quantum nuclear effects into consideration for the transition between two states. While free-energy profiles typically exhibit similarities to those in Fig. 4c, there are some deviations in local minimums that warrant further discussion. The inclusion of a supporting video could enhance the presentation.

- 5) Can a stable region of the novel "flat-rhombic" ice be identified, or is it a meta-stable phase? Furthermore, simulation packages typically calculate stress tensor elements over the entire simulation box. To determine the effective lateral pressure applied to the monolayer ice, a proper conversion is necessary, as suggested in *Phys. Rev. Lett.* 2016, 116, 025501.
- 6) In this manuscript, a Morse potential was employed to mimic the confining potential from the interaction between water and carbon. It is advisable for the author to provide the fitting parameters, aiding readers in reproducing their findings.
- 7) The confining effect was simulated with a pseudo-wall and no explicit lattice information was introduced. It will be better to further investigate the stability of the "flat-rhombic" phase confined by real carbons (i.e., graphene).
- 8) The author analyzed the orientation of water molecules and ascribed the distinctive properties of the "flat-rhombic" structure to the long-range ordered arrangement of molecular orientations and dipoles. However, the analysis of orientation neglected the nuclear quantum effect (NQE). Considering NQE implies that hydrogen atoms move akin to an electron cloud within a specific spatial range. Consequently, the orientations might not be predictable values but rather a cloud-like set.
- 9) Additional validation and testing should be conducted to ensure the accuracy of the machine learning potential. It is crucial to verify whether the new phase predicted in this study is included in the training data, as machine learning potentials are generally proficient at interpolation but may face challenges with extrapolation.

Reviewer #3 (Remarks to the Author):

Reviewer #3 Attachment on the following page

2D water/ice has received considerable interests over the past few years. Recent high-profile publications in Nature (sister) journals (e.g., Kapil et. al., Nature 2022 (ref. 17) and Jiang et al., Nature Physics (Jan. 22, 2024)) have shown that 2D water/ice in nanoconfinement entails rich phase behaviour and proton dynamical behaviour that differ from those of bulk ices. The manuscript titled "A Quasi-One-Dimensional Hydrogen-Bonded Monolayer Ice Phase" by Ravindra et al. presents a comprehensive computational study of confined monolayer ice that exhibits marked deviation from the well-known ice rule and the intriguing coherent proton dynamics in the "flat-rhombic" phase of 2D water. This topic, as well as machine-learning based simulation work, are certainly of broad interests. I would recommend publication of this work in Nature Communications after the authors address the following technical comments.

Technical comments:

- (1) In an article (JACS 2021, Ref. 16), Jiang et al. reported ab initio MD simulation (with smaller systems than this work) study of 2D water/ice in nanoconfinement, and found a flat zig-zag monolayer ice (ZZMI) phase that also does not obey the ice rule. So, it is important to let the readers know the difference and relation between the ZZMI phase reported in JACS 2021 (Ref. 16) and the "flat-rhombic" phase here in structure, hydrogen-bonding pattern, their location in the T-P phase diagram (Nature 2022), and their dependence on the nanoslit width. A comprehensive study of both phases would be highly informative to the water/ice community.
- (2) In the abstract, the authors gave the statement about "setting the stage for exploiting electronic, vibronic, and optical properties" of the "flat-rhombic" phase. In the main paper, however, none of these properties have been investigated or discussed. The authors should provide some computational results of these properties to be consistent with the Abstract, and to increase the novelty of this work for publication in Nature Communications.
- (3) In Table I, the energy analysis, although valid for hexagonal and flat-rhombic phases, may be inapplicable to the pentagonal phase, as the selected water molecule chain alone may be unstable.
- (4) On the analysis of proton dynamics in the flat-rhombic phase at the point of melting into the hexatic phase, could the dynamics reflect a pre-melting behaviour? Clarification on this point is needed.
- (5) An explanation on the observed additional 0.2 hydrogen bonds in flat-rhombic ice at low temperatures (20 K) from PIMD simulation (compared to AIMD) should be given. This deviation is also present in hexagonal and pentagonal phases (see Supplementary Figure 3), suggesting more hydrogen bonds formed due to quantum nuclear effects. This interesting phenomenon should be explored and discussed further.

- (6) The authors should show that the training data for PIMD simulations are sufficient to model the quantum nuclear effect.
- (7) The proton disorder should be analyzed, for example, the net dipole of each water chain is not aligned at the initial state.
- (8) Finally, the ML simulation results presented in this work should be verified through benchmark ab initio simulations to ensure the robustness of the findings.

Reviewer #4 (Remarks to the Author):

In this study, the authors use atomistic simulations based on a machine learning potential (trained with first-principles data), to analyze the hydrogen bond structure and elucidate the origin of the interactions that stabilize an unusual ice phase within a water monolayer under hydrophobic confinement. I enjoyed reading this manuscript, which is technically sound and presents interesting findings. Certain aspects of the results discussion were a bit confusing, however, and the flow of the manuscript along with some figures could benefit from improvements for greater clarity.

I am not convinced that the main original findings will appeal to a broad audience, as the flat rhombic ice phase was identified in previous work, and deviations from the Bernal-Fowler ice rules are not uncommon. I consider the possibility of concerted proton dynamics the most interesting contribution in this work, however the data presented in the paper does not demonstrate such behavior. Moreover, I am concerned with possible artifacts in the simulations arising from finite size effects. In an effort to improve the narrative appeal, the authors suggest that the ice phase under study may fall into the category of quasi-one-dimensional vdW materials. I am doubtful, however, that this ice phase is stable under more realistic nanoconfinement conditions (i.e., conditions that deviate from the idealized perfectly smooth, perfectly rigid walls used in the study), even in simulations. Here are my comments below, which I hope will contribute to enhance the manuscript:

Comments:

- The use of the term 'graphene-like nanocavity' (along with other references to graphene) seems somewhat misleading, considering that the hydrophobic confinement in the simulations is created by perfectly rigid, smooth hydrophobic walls (i.e., the potential is 'uniform in the plane of confinement'). While the Morse potential was parameterized based on water-carbon interactions, the specific arrangement of carbon atoms in the hydrophobic wall would likely have a crucial impact on the stability of the ice phases at this level of confinement.
- On a related note, I wonder if a perfectly rigid and perfectly smooth wall is required to stabilize this unusual ice phase. I maintain certain skepticism about whether this phase would be stable when a real hydrophobic material (e.g., graphene) is used as confining wall. For instance, the out-of-plane deformations of the confining surfaces have been shown to play an important role in the structure of nanoconfined water (e.g., ACS Nano 2018, 12, 1, 448–454; Sci Rep 2017, 7, 2646). The authors should address whether the realistic confinement conditions detailed in those, and similar studies would impact their findings.
- I question whether the lateral pressures simulated (e.g., 2 GPa) align with what could realistically be generated by 'vdW confinement'. Ultimately, if there is a possibility of observing this phase in laboratory conditions, it is essential we understand the specific conditions under which it could be expected to occur.
- The authors' use of italics seems stylistically inconsistent. At times, italics are used for emphasis, such as in 'just two hydrogen bonds per water molecule,' and at other times, seemingly for qualifiers, like 'covalently bonded.' A more consistent application of italics for specific purposes would enhance the readability.
- There is a typo in the caption of Table I, it mentions '... the lattice energy, Echain,' which should correctly be 'Elattice.'
- The discussion involving percentages in the right column of Page 3 is somewhat confusing, primarily because the relationship between the different 'types' of binding energies selected by the authors isn't immediately clear. Including at least the equation $E_{stack} = E_{lattice} - E_{chain}$ from the methods into the results section—or possibly the entire relevant paragraph—would significantly improve the paper's flow.
- I believe the characterization of the hexagonal ice phase as having only a single chain motif is inaccurate. Similar to graphene, there should be 'zigzag' and 'armchair' directions, which would

imply the existence of two distinct chain types.

- The legend in Figure 1b might be misleading or possibly incorrect. It's unclear whether the fully colored section of the bars represents the 'Total' or 'Total - vdW' energy. I assume it's the latter. If so, please consider relabeling for clarity.

- A clear and precise definition of the angles theta and phi is missing in the text. I find the graphical illustration in Figure 2a confusing, and the 'intuitive' explanation of their meaning in the caption of Figure 2 unhelpful. Please, specify the vector and the plane against which these angles are defined.

- I find Figure 2a quite confusing. The stability of a given configuration is determined within a specific region of the (theta, phi) space, yet only one angle is depicted in each panel. It would also be helpful to clarify the precise criteria used for determining stability.

- I understand that the angle phi is periodic, with a period of $\pi/2$. Could you explain why the plots extend from $-\pi/2$ to $\pi/2$ (instead of just 0 to $\pi/2$)?

- The labels in Fig 2b for the states illustrated in Fig 2a obscure the data. While I question the necessity of highlighting these states, if the authors choose to include them, an alternative presentation should be sought to avoid hiding the data.

- To enhance the clarity and flow of the paper, it might be beneficial to include a separate panel for the number of putative hydrogen bonds shown in Figure 2b, right column.

- There is a crucial discussion about the connection between regions of low probability and a higher number of putative hydrogen bonds. However, the data in Figure 2 presents $\log(p)$ and the number of hydrogen bonds independently, as functions of phi and theta. The manuscript would greatly benefit from a density plot illustrating the correlation between $\log(p)$ and the number of hydrogen bonds. Without this, assessing the validity of the authors' observations becomes challenging.

- The authors state, 'the peculiar hydrogen bonding behavior [...] emphasizes the importance of incorporating dynamical information into the hydrogen bond definition.' However, from my understanding, the authors do not undertake such an approach but rather adhere to the geometric 'canonical' definition used by Chandler. Could you clarify the rationale behind emphasizing the need for dynamical information in the hydrogen bond definition?

- The colors used in Figure 3b are not easy to distinguish, and the symbols are the same for all the series.

- The authors are encouraged to show the error bars in the calculation of the h-bond lifetime presented in Fig 3c.

- A considerable portion of the text is dedicated to discussing the dielectric constant of the ice phase, yet the relevant figure is only included in the SI. It might be more appropriate to feature this figure in the main text, or alternatively, consolidate the discussion about it.

- The authors transition from discussing findings in a system of 576 molecules to one with 144 molecules, justifying this shift to the observation of concerted dynamics in the smaller systems. If such dynamics are not observed in the larger system, should there not be concerns regarding artifacts originated from finite size effects? This aspect appears troubling.

- The presentation of the data in Figure 4b makes the behavior it intends to reveal unclear. The authors are encouraged to consider a different style of plotting, such as stacked series, to ensure the time evolution is clearly discernible across the three temperatures.

- The findings depicted in Figure 4 appear very intuitive but their connection to coherent proton dynamics is not immediately clear to me. Wouldn't an analysis of dynamical correlations would be necessary to assess this connection?

- The authors refer to the 'observed length scale of concerted motion' in the last paragraph of the Results section, but I find this confusing as there is no prior discussion or presentation of results related to length scales in the manuscript.

Find below point-by-point responses to the reviewers' comments (in red color), immediately followed by the changes made to the manuscript (in blue color). In addition to the updated manuscript file, we also provide a separate manuscript file with the changes highlighted in blue color.

Reviewer #1 (Remarks to the Author):

In this manuscript, the author introduces a novel "flat-rhombic" ice characterized by one-dimensional water chains, with van der Waals interactions dominating the inter-chain interactions. Notably, each water molecule in this quasi-one-dimensional hydrogen-bonded monolayer ice forms only two hydrogen bonds, and the prevalence of van der Waals interactions leads to a deviation from the conventional ice rule. Additionally, the author reports a concerted motion behavior within the hydrogen bonding network of the "flat-rhombic" ice. These findings hold fundamental significance in water science and have widespread interest among researchers. However, certain critical issues need to be addressed before considering this manuscript for publication in Nat. Commun.

We thank the reviewer for their positive comments and for acknowledging the fundamental significance of the key results for researchers in the water science community and beyond.

1) A more comprehensive discussion is needed to clarify the distinctions between the quasi-one-dimensional monolayer ice and the flat-rhombic phase presented by Kapil et al in Ref. 17, as well as the zigzag quasi-bilayer ice reported in Ref. 18. Particularly, it seems that the latter exhibits some critical similarities with the "flat-rhombic" ice reported here. For instance, both configurations consist of one-dimensional water chains connected through H-bonding, subsequently stacking together through van der Waals (vdW) forces (it is not mentioned in Ref. 18). Additionally, both ices involve water molecules forming only two hydrogen bonds. Is the "flat-rhombic" ice considered a special case existing exclusively within the 5 Å nanoslit?

We thank the reviewer for raising this point. While we focused on only the flat rhombic phase from Ref. 17, the phenomena of vdW interactions stabilizing stacks of one-dimensional zig-zagging water chains is also seen in other confined ice phases from Ref. 18, as correctly pointed by the reviewer.

1. The monolayer one-dimensional ice considered in this work corresponds to the flat rhombic phase from Ref. 17 stabilized in a graphene nano capillary of 5 Angstrom width.

2. This phase is nearly the same as the zig-zagging quasi-monolayer ice (zzQMI) phase from Ref. 18. The key difference is the small buckling of zzQMI due to a higher confinement width of 6.5 Angstrom.

3. As correctly pointed out by the reviewer, the zig-zagging quasi-bilayer ice (zzQBI) from Ref. 18 consists of hydrogen bonded chains as discussed for the above phases. The main difference is that the zzQBI

comprises a bilayer of one-dimensional chains. It is stable at high confining pressures.

Taking the reviewer's comments into account, we have incorporated a more comprehensive discussion of the known ice phases (see below). We clearly highlight all known confined ice phases comprising zig-zagging hydrogen-bonded chains and discuss our results in the context of confined ice rather than just the "flat-rhombic" phase from Ref. 17.

We have added the following to the introduction of the main text:

"More recently, Lin et al. [17] extended this investigation to larger confinement widths and a wider range of confinement pressures, revealing new bilayer ice phases."

and the following sentences to section III.A

"Lin et al. [17] explored even higher lateral pressures for a confinement width of 6.5 Å and confirmed the existence of a zigzagging monolayer ice (ZZMI) beyond 0.5 GPa. This phase is topologically the same as the flat-rhombic phase, except that the larger confinement width leads to a small buckling of the oxygen atoms. In the 15-20 GPa regime, they report a zigzagging quasi bilayer ice (ZZ-qBI) structure as the stable phase. This phase also resembles the flat-rhombic phase, although the buckling in this phase results in two nearly distinct layers of water molecules."

We have also included new calculations to understand the role of van der Waals (vdW) interactions in stabilizing the bilayer zig-zagging phase. We find that this phase also exhibits quasi-one-dimensional hydrogen bonding. The hydrogen-bonded chains provide most of the stabilization energy, while the vdW interactions mainly contribute to the stacking energy of the chains. We also observed that without the vdW stacking, this phase has a positive lattice energy, which means that it would be thermodynamically unstable.

"Similarly, the stacking energy of the ZZ-qBI phase is almost entirely driven by vdW interactions, meaning that vdW forces also stabilize the interactions between this phase's hydrogen-bonded chains. In fact, in the absence of vdW stacking, the lattice energy is positive, meaning that the ZZ-qBI crystal structure would be thermodynamically unstable. This suggests that these vdW interactions between hydrogen-bonded chains play an essential role in stabilizing nanoconfined crystal structures at high lateral pressures."

These additions strengthen our conclusions. One-dimensional hydrogen bonding is not limited to flat-rhombic ice but extends to other phases across confinement conditions. Crucially, this phenomenon becomes increasingly essential in high-pressure confinement conditions.

2) The author claimed that the van der Waals interactions play a critical role in stabilizing the "flat-rhombic" ice. The definitions of stabilization energies lack clarity. Should $E_{\text{vdW}} = E_{\text{vdW}} -$

E^*) be understood as $E_{\text{relaxed}} = E_{\text{single-point}} - N * E^*$, where N is number of water molecules in a unitcell? Is $E_{\text{single-point}}$ the energy of the fully relaxed unit cell? Typically, the term "single-point energy" refers to the energy of the structure as it is built without any relaxation performed. Furthermore, to enhance the robustness of the conclusion, it would be beneficial to present an additional example where van der Waals interactions serve as the primary stacking force. The zigzag quasi-bilayer ice mentioned in Q#1 could serve as a potential candidate for such an illustration.

We thank the reviewer for flagging this. We agree that our previous notation may have led to some confusion. We now clearly provide details of the calculations. The initial structures of the confined ice phases correspond to geometry-optimised structures from Ref. 18 to perform a meaningful static thermodynamic stability calculation. To estimate E_{crystal} and E_{chain} , we carve our chains and molecules from the crystalline structure and perform "single point energy" calculations without optimising the chains and molecules.

"We perform these calculations on geometry-optimized structures of the confined ice phases [17]. The geometry-optimized structures of the hexagonal, pentagonal, and flat-rhombic monolayer phases were taken from Ref. 16. In contrast, the structure of the zigzagging quasi bilayer ice from Ref. 17 was optimized using the CP2K code [64]. We define the stabilization energy of a crystal with respect to an isolated molecule as $E_{\text{lattice}} = E_{\text{crystal}} - E_{\text{gas}}$, where E_{crystal} is the single-point energy of the crystalline lattice per water molecule, and E_{gas} is the single-point energy of an isolated water molecule in vacuum. Similarly, we define the stabilization energy of a chain of water molecules with respect to an isolated molecule as $E_{\text{chain}} = E_{\text{struct. chain}} - E_{\text{gas}}$, where $E_{\text{struct. chain}}$ is the single-point energy of the chain structure, and the stabilization energy of a crystal with respect to a chain of water molecules as $E_{\text{stack}} = E_{\text{lattice}} - E_{\text{chain}}$."

As seen above, we have extended our analysis to the bilayer zig-zagging ice phase. We take the initial structure from Ref. 18 and perform a fixed-cell geometry optimization at the revPBE0- D3 level of theory with CP2K. As explained above, the E_{crystal} and E_{chain} calculations are single-point energy calculations. We include the new results in Table 1 and Fig 1.

Phase	Geometric hydrogen bonds			E_{lattice} (kcal/mol)		E_{chain} (kcal/mol)		E_{stack} (kcal/mol)	
	donors	acceptors	total	total	vdW	total	vdW	total	vdW
Hexagonal	1.5	1.5	3.0	-9.05	-1.79	-5.12 ± 0.31	-0.85 ± 0.17	-3.93 ± 0.31	-0.94 ± 0.17
Pentagonal	1.6	1.6	3.2	-8.99	-2.24	-3.20 ± 0.50	-1.10 ± 0.50	-5.80 ± 0.50	-1.10 ± 0.50
Flat-rhombic	1.0	1.0	2.0	-8.75	-2.76	-6.76	-1.08	-2.00	-1.68
ZZ-qBI	1.0	1.0	2.0	-12.71	-19.91	-9.69	-4.65	-3.05	-15.33

We thank the reviewer for this suggestion, as it has strengthened our conclusions. By analysing the role of vdW interactions in the bilayer zig-zagging ice phase, we found that this phase can also be classified as a quasi-one-dimensional hydrogen-bonded phase. Moreover, this phase is thermodynamically unstable in the absence of vdW interactions, highlighting the importance of quasi-one-dimensional hydrogen bonding at high confinement pressures.

3) The author has described an interesting, concerted motion behavior within the hydrogen bonding network of the "flat-rhombic" ice in the system consisting of 144 molecules. Is this behavior dependent on the system size, and will it dissipate in larger systems?

We thank the reviewer for this comment. This effect is not observed in the timescales of the simulations performed with 576 molecules, suggesting that the free energy barrier for concerted motion is size-extensive. In a much larger system, we expect to either observe an ordered phase or the formation of motifs exhibiting this behaviour. To make this point clear, we have included the following in the main text:

“As stated earlier, this concerted motion is not observed in the larger simulation cell size that contains 576 molecules for the timescales we consider: it is only seen in simulations with 144 water molecules. This suggests that the free energy barrier for this concerted motion is size-extensive. As a result, we would expect a larger system size to either exhibit an ordered phase or a phase in which smaller domains that exhibit this behavior coexist.”

Nonetheless, our results for 144 molecules remain significant as we are not aiming to understand coherent proton disorder in a thermodynamically large system. Instead, we are focusing on a specific situation when our simulation cells correspond to sizes of nanocavities that are experimentally

accessible. In particular, the length scale considered in our simulations corresponds to nanopockets of a few nanometers in dimensions when water is encapsulated between graphene sheets. We have made this point more evident in the main text:

“However, crystallites of nanoconfined water molecules containing around 100 water molecules are experimentally accessible, so such concerted motion could be observed in experiments as small ice crystallites forming between graphene sheets [46] or high-pressure graphene nanobubbles formed by irradiation [47]. Previous work has suggested that the simulation setup we employ is still reasonable for studying such encapsulated systems, even without explicit confining atoms [46, 48].”

4) Regarding the concerted motion mentioned in Q#3, the author computed the free-energy variation with the σ parameter. It would be beneficial to include additional details on the methodology employed to calculate the free-energy profiles depicted in Fig. 4c. The author also took quantum nuclear effects into consideration for the transition between two states. While free-energy profiles typically exhibit similarities to those in Fig. 4c, there are some deviations in local minimums that warrant further discussion. The inclusion of a supporting video could enhance the presentation.

We appreciate this feedback. We have included a note in the supporting information detailing the calculation of the free energy profiles.

“We used a simple binning procedure to compute free energy profiles along a particular variable, x . Given a molecular dynamics trajectory, we assemble a list of all values that the variable x takes on in each frame of the trajectory. We then compute the probability histogram $P(x)$ with an appropriately chosen number of bins. The free energy profile in units of $k_B T$ is then reported as $F(x) = -\log(P(x))$, where k_B is Boltzmann’s constant.”

Further we have expanded upon the differences in the free energy profiles due to quantum nuclear effects.

“Furthermore, since nuclear quantum effects lead to increased proton disorder, the σ parameter gets pushed closer to 0 on average. This causes the minima in the free energy profiles along σ to move closer to 0 in our PIMD simulations as compared to the free energy profiles from our classical simulations.”

We have also included a video of the concerted hydrogen bond dynamics.

5) Can a stable region of the novel "flat-rhombic" ice be identified, or is it a meta-stable phase? Furthermore, simulation packages typically calculate stress tensor elements over the entire simulation box. To determine the effective lateral pressure applied to the monolayer ice, a proper conversion is necessary, as suggested in Phys. Rev. Lett. 2016, 116, 025501.

We thank the reviewer for this comment. We hope that our response to the first comment clarifies this confusion, highlighting that the flat rhombic phase is not novel.

However, to answer the reviewer, the flat rhombic phase can be found in the temperature - (lateral) pressure phase diagram of Ref. 17. This phase is thermodynamically stable at lateral pressure 0.5 - 4.0 GPa and temperatures 0 - 320 K. In our earlier work, we estimated the later pressure correctly, as suggested in the above reference.

6) In this manuscript, a Morse potential was employed to mimic the confining potential from the interaction between water and carbon. It is advisable for the author to provide the fitting parameters, aiding readers in reproducing their findings.

We agree with the reviewer and have included the parameters of the confining potential in the Appendix (Section A5).

7) The confining effect was simulated with a pseudo-wall and no explicit lattice information was introduced. It will be better to further investigate the stability of the "flat-rhombic" phase confined by real carbons (i.e., graphene)..

We appreciate the reviewer's insightful comment and recognize the significance of further investigating the stability of the "flat-rhombic" phase when confined by real carbon. However, we wish to highlight the fundamental importance of our results obtained using an "atomistically flat" confining potential. Such a potential allows understanding topological confinement effects on water — a phenomenon that extends beyond the specific context of water-carbon interactions. The simplicity of an atomistically flat potential allow a cleaner interpretation of these effects, which might get obfuscated by the role of interactions in real materials.

We agree with the general interest in the real-material effects on the stability of confined ice phases. Such an investigation will require significantly higher level of complexity due to both sampling issues and the lack of high-end electronic structure benchmarks on graphene confined ice. However, we are able to inform the reviewers in good faith that preliminary results from ongoing work indeed suggest that the flat-rhombic phase maintains its stability in the presence of real carbon atoms. We have emphasised this point in the main text.

“Such a potential, characterized by perfectly smooth walls, has been widely adopted in previous studies, both in force field [9–11] and first-principles research [12, 14–17, 59]. The uniform confinement potential has demonstrated semi-quantitative accuracy in describing the behavior of water confined within graphene-like cavities [11], as corroborated by a good agreement of stable phases and melting temperatures with respect to confinement simulations that include explicit carbon atoms [48]. Furthermore, the uniform confinement model’s atomistically flat nature allows for a clean interpretation of topological confinement effects – a phenomenon that extends beyond the specific context of graphene-based confinement.”

8) The author analyzed the orientation of water molecules and ascribed the distinctive properties of the "flat-rhombic" structure to the long-range ordered arrangement of molecular orientations and dipoles. However, the analysis of orientation neglected the nuclear quantum effect (NQE). Considering NQE implies that hydrogen atoms move akin to an electron cloud within a specific spatial range. Consequently, the orientations might not be predictable values but rather a cloud-like set.

We thank the reviewer for this comment. While the instantaneous orientations may not be predictable the thermodynamic average, as measured in experiments, remains well defined. We believe this confusion arose due to lack of detailed technical information on the calculation of probability distributions and free energy surfaces. To address this we have detailed the calculation of the histograms.

"For structural and thermodynamic quantities (e.g., free energy profiles), we average results over all the imaginary time slices from the PIMD simulations."

9) Additional validation and testing should be conducted to ensure the accuracy of the machine learning potential. It is crucial to verify whether the new phase predicted in this study is included in the training data, as machine learning potentials are generally proficient at interpolation but may face challenges with Extrapolation.

We agree with the reviewer that a thorough error analysis would be extremely useful, so we have included the energy/force errors for our system across a range of conditions. The tabulated errors are in Table II, and a description of how these errors were computed was added to the appendix.

"To assess the neural network's ability to generalize to unseen configurations, we compared the energies and forces produced by the MLP to those produced by the reference revPBE0-D3 functional on new structures that were not seen during training time. These structures were generated by running 2 nanosecond NPxyT simulations at 2.0 GPa starting from each of the solid phases identified in Ref. 16. Each solid phase was simulated at 100 K, 300 K, and 600 K. For some of these simulations, the solid phase was metastable, and so the trajectory remained in the solid phase. For other simulations, the solid phase was unstable, and the water molecules in the simulation entered a disordered phase or glassy phase (depending on the conditions). Hence, these trajectories contained a diverse set of ordered and disordered configurations that span a broad range of local molecular configurations. We computed the errors between the MLP's predictions and the reference revPBE0-D3 calculations over all of the configurations generated in each of these trajectories. The resulting mean absolute errors (MAE) for the energies and root-mean-squared errors (RMSE) for average atomic forces are shown in Supplementary Table I. These errors are similar in magnitude to the errors of MLPs employed in other studies [16, 65, 66]."

Reviewer # 2 (Remarks to the Author):

2D water/ice has received considerable interests over the past few years. Recent high-profile publications in Nature (sister) journals (e.g., Kapil et. al., Nature 2022 (ref. 17) and Jiang et al., Nature Physics (Jan. 22, 2024)) have shown that 2D water/ice in nanoconfinement entails rich phase behaviour and proton dynamical behaviour that differ from those of bulk ices. The manuscript titled "A Quasi-One-Dimensional Hydrogen-Bonded Monolayer Ice Phase" by Ravindra et al. presents a comprehensive computational study of confined monolayer ice that exhibits marked deviation from the well-known ice rule and the intriguing coherent proton dynamics in the "flat-rhombic" phase of 2D water. This topic, as well as machine-learning based simulation work, are certainly of broad interests. I would recommend publication of this work in Nature Communications after the authors address the following technical comments.

We thank the reviewer for the overall positive comments on the methodology and the results.

Technical comments:

(1) In an article (JACS 2021, Ref. 16), Jiang et al. reported ab initio MD simulation (with smaller systems than this work) study of 2D water/ice in nanoconfinement, and found a flat zig-zag monolayer ice (ZZMI) phase that also does not obey the ice rule. So, it is important to let the readers know the difference and relation between the ZZMI phase reported in JACS 2021 (Ref. 16) and the "flat-rhombic" phase here in structure, hydrogen-bonding pattern, their location in the T-P phase diagram (Nature 2022), and their dependence on the nanoslit width. A comprehensive study of both phases would be highly informative to the water/ice community.

We thank the reviewer for this suggestion. It would be helpful to extend the scope of this manuscript to other confined ice phases.

Indeed, Ref.16 reports the zig-zagging monolayer ice phase (ZZMI). However, it is the same phase as "flat-rhombic" ice from Ref. 17. The only difference is the slight but noticeable buckling of the oxygen atoms arising from the increase in the width of the confining slit (from 5 Å in Ref.17 to 6.5 Å in Ref. 16). Hence, "flat-rhombic" ice and ZZMI correspond to the same phase in different conditions. While we don't repeat our calculations for this phase, we clearly state the distinction and similarities between these phases and discuss our results in the context of both of these phases rather than just the flat rhombic phase:

"Lin et al. [17] explored even higher lateral pressures for a confinement width of 6.5 Å and confirmed the existence of a zigzagging monolayer ice (ZZMI) beyond 0.5 GPa. This phase is topologically the same as the flat-rhombic phase, except that the larger confinement width leads to a small buckling of the oxygen atoms. In the 15-20 GPa regime, they report a zigzagging quasi bilayer ice (ZZ-qBI) structure as

the stable phase. This phase also resembles the flat-rhombic phase, although the buckling in this phase results in two nearly distinct layers of water molecules.”

Similarly, following the suggestions of reviewer #1, we have extended our analysis to the ZZqBI —an intriguing high-pressure bilayer equivalent of the flat-rhombic or ZZMI phase. We are pleased to note that our calculations confirm that vdW stacking of strongly hydrogen-bonded chains is the primary stabilization mechanism in ZZqBI. Crucially, without vdW corrections, this high-density phase is thermodynamically unstable. Find below the changes to the manuscript:

“Similarly, the stacking energy of the ZZ-qBI phase is almost entirely driven by vdW interactions, meaning that vdW forces also stabilize the interactions between this phase’s hydrogen-bonded chains. In fact, in the absence of vdW stacking, the lattice energy is positive, meaning that the ZZ-qBI crystal structure would be thermodynamically unstable. This suggests that these vdW interactions between hydrogen-bonded chains play an essential role in stabilizing nanoconfined crystal structures at high lateral pressures.”

(2) In the abstract, the authors gave the statement about “setting the stage for exploiting electronic, vibronic, and optical properties” of the “flat-rhombic” phase. In the main paper, however, none of these properties have been investigated or discussed. The authors should provide some computational results of these properties to be consistent with the Abstract, and to increase the novelty of this work for publication in Nature Communications.

We thank the reviewer for their comment. We intended to suggest other avenues that our work may open up, but we agree that it may seem empty in the absence of additional results. To strengthen the case, and in keeping with reviewer #3’s suggestion, we have moved the section and graphics discussing the dielectric response and potential ferroelectricity of the flat rhombic phase to the main text. As can be seen in Fig. 5, our results are in agreement with the in-plane experimental dielectric responses.

We have also relaxed our language on the general electronic, vibrational and electronic properties of ice as a complete analysis goes beyond the scope of this publication, where we have focused on identifying an interesting stabilisation mechanism. Instead, in the abstract, we now focus on the properties and observables studied in this work

“The unusual interplay of hydrogen bonding and van der Waals interactions in nanoconfined ice results in atypical proton behavior such as potential ferroelectric behavior, low dielectric response, and long-range proton dynamics.”

In addition with our new results on the ZZQBI phase, we have made a strong case that the phenomena of quasi-one-dimensional hydrogen bonding are applicable to different ice phases and confinement conditions. In the abstract, we now discuss these results more generally in the context of nanoconfined ice, as opposed to the flat rhombic phase in the previous iteration

“In this study, we employ machine learning-driven first-principles simulations to identify a new stabilization mechanism in nanoconfined ice phases beyond conventional ice rules. Instead of forming four hydrogen bonds, nanoconfined crystalline ice can form a quasi-one-dimensional hydrogen-bonded structure that exhibits only two hydrogen bonds per water molecule.”

(3) In Table I, the energy analysis, although valid for hexagonal and flat-rhombic phases, may be inapplicable to the pentagonal phase, as the selected water molecule chain alone may be unstable.

We thank the reviewer for this comment. Indeed, the structure of the pentagonal phase has many hydrogen-bonded motifs, meaning that it isn't very instructive to perform such a single-chain analysis. Due to this reason, we mentioned in the text that we averaged these results over several chains to get a general idea of the relative role of hydrogen bonding in contributing to the stabilisation energy of each nanoconfined phase. To ensure readers do not miss this point, we have added the following sentences.

“As can be seen in the pentagonal phase snapshot in Fig. 1(a), there is no clear choice for a unique hydrogen bond chain in the pentagonal phase. This shows qualitatively that the pentagonal phase cannot clearly be separated into distinct hydrogen bond chains. Nonetheless, to perform the quantitative comparison shown in Fig. 1(b), we averaged our energies over three different hydrogen bond chains in the pentagonal phase. The energies for these different chains were all extremely close to each other, suggesting that this quantitative comparison is robust with respect to the exact choice of hydrogen bond chain in the pentagonal phase.”

(4) On the analysis of proton dynamics in the flat-rhombic phase at the point of melting into the hexatic phase, could the dynamics reflect a pre-melting behaviour? Clarification on this point is needed.

We thank the reviewer for this intriguing comment. As characterised in Fig. 3(a), the flat rhombic phase exhibits peculiar librational motion of water molecules, similar to that of a liquid. However, given that the distribution functions at 380 K and 280 K are qualitatively the same, we believe our trajectories at 380 K correspond to the metastable phase space associated with the solid phase. The free energy profiles in Fig. 6(a) behave similarly. Therefore, we do not think the disorder observed is a pre-melting behaviour. We have clarified this point in the main text.

“While the highest temperatures explored here are greater than the melting temperature of ice, we believe that our trajectories correspond to a metastable state associated with the solid phase. Since the free energy profiles at the intermediate and high-temperature conditions are qualitatively the same, we think that the near-free exploration of molecular configurations at 380 K is a result of thermal activation, rather than premelting behavior.”

(5) An explanation on the observed additional 0.2 hydrogen bonds in flat-rhombic ice at low temperatures (20 K) from PIMD simulation (compared to AIMD) should be given. This deviation is also present in hexagonal and pentagonal phases (see Supplementary Figure 3), suggesting more hydrogen bonds formed due to quantum nuclear effects. This interesting phenomenon should be explored and

discussed further

We agree that this observation is interesting, so we have elaborated on this point in the main text, highlighting that the additional 0.2 hydrogen bonds are only geometric. They do not last long enough to be considered real hydrogen bonds. To make this point clear, we have also included data corresponding to the number of real hydrogen bonds by incorporating dynamical information into the definition of a hydrogen bond.

“In the same figure, we also observe that nuclear quantum effects apparently exhibit an additional 0.2 putative hydrogen bonds, even at temperatures as low as 20 K. This is the result of the zero-point fluctuations lowering the energy barrier for sampling these additional hydrogen-bonded configurations, as we elaborate upon in Section V of the Supplementary Information.”

(6) The authors should show that the training data for PIMD simulations are sufficient to model the quantum nuclear effect.

We thank the reviewer for this comment. We have now clarified that we used PIMD data to train the model, so the model isn't "extrapolating" when accounting for quantum nuclear effects. In addition, we have provided a characterisation of the error and force energies of our force field in the SM. This error analysis considers the broad range of lateral pressure and temperature conditions that we consider in this work, including the new configurations explored by the inclusion of nuclear quantum effects.

“We also included structures from PIMD simulation trajectories so that the training dataset includes the new configurations that are accessible with the inclusion of nuclear quantum effects.”

(7) The proton disorder should be analyzed, for example, the net dipole of each water chain is not aligned at the initial state.

We have gone ahead and added a figure in the main text (Figure 2) to emphasise this point, as well as provided a better depiction of the theta and phi angle definitions in this work. We have also moved our formal definitions of the phi, theta, and sigma parameters into the main text and elaborated on their definitions. We hope that this provides a more thorough context for our analysis on proton disorder in Figs. 5 and 6.

(8) Finally, the ML simulation results presented in this work should be verified through benchmark ab initio simulations to ensure the robustness of the findings.

We fully agree with this point. We have provided a characterisation of the error and force energies of our force field in the SM. This error analysis considers the broad range of conditions that we consider in this work, including the new configurations explored by the inclusion of nuclear quantum effects. The tabulated errors are in Table II, and a description of how these errors were computed was added to the appendix.

“To assess the neural network’s ability to generalize to unseen configurations, we compared the energies and forces produced by the MLP to those produced by the reference revPBE0-D3 functional on new structures that were not seen during training time. These structures were generated by running 2 nanosecond NPxyT simulations at 2.0 GPa starting from each of the solid phases identified in Ref. 16. Each solid phase was simulated at 100 K, 300 K, and 600 K. For some of these simulations, the solid phase was metastable, and so the trajectory remained in the solid phase. For other simulations, the solid phase was unstable, and the water molecules in the simulation entered a disordered phase or glassy phase (depending on the conditions). Hence, these trajectories contained a diverse set of ordered and disordered configurations that span a broad range of local molecular configurations. We computed the errors between the MLP’s predictions and the reference revPBE0-D3 calculations over all of the configurations generated in each of these trajectories. The resulting mean absolute errors (MAE) for the energies and root-mean-squared errors (RMSE) for average atomic forces are shown in Supplementary Table I. These errors are similar in magnitude to the errors of MLPs employed in other studies [16, 65, 66].”

Reviewer #4 (Remarks to the Author):

In this study, the authors use atomistic simulations based on a machine learning potential (trained with first-principles data), to analyze the hydrogen bond structure and elucidate the origin of the interactions that stabilize an unusual ice phase within a water monolayer under hydrophobic confinement. I enjoyed reading this manuscript, which is technically sound and presents interesting findings. Certain aspects of the results discussion were a bit confusing, however, and the flow of the manuscript along with some figures could benefit from improvements for greater clarity.

I am not convinced that the main original findings will appeal to a broad audience, as the flat rhombic ice phase was identified in previous work, and deviations from the Bernal-Fowler ice rules are not uncommon. I consider the possibility of concerted proton dynamics the most interesting contribution in this work, however the data presented in the paper does not demonstrate such behavior. Moreover, I am concerned with possible artifacts in the simulations arising from finite size effects. In an effort to improve the narrative appeal, the authors suggest that the ice phase under study may fall into the category of quasi-one-dimensional vdW materials. I am doubtful, however, that this ice phase is stable under more realistic nanoconfinement conditions (i.e., conditions that deviate from the idealized perfectly smooth, perfectly rigid walls used in the study), even in simulations. Here are my comments below, which I hope will contribute to enhance the manuscript:

We thank the referee for their constructive criticism. We want to emphasize a few key points before responding to their comments.

1. We agree that there exist exceptions to Bernal-Fowler ice rules. However, the main message of the paper is not that ice rules are violated in nanoscale confinement. Instead, the novelty here is that we find vdW interactions stabilize one dimensionally hydrogen-bonded motifs. Thanks to suggestions by referees #1-#2, we have now shown that the presented mechanism also holds for different confinement widths and phases. In fact, without vdW interactions, the zig-zagging bilayer phase would be thermodynamically unstable.

2. We agree that finite size effects are always a concern when exploring coherent motion of atoms. However, we are not aiming to understand coherent proton disorder in a thermodynamically large system. Instead, we are focusing on a specific situation when our simulation cells correspond to sizes of nanocavities that are experimentally accessible.

3. Finally, uniform confining potential to understand the general properties of ice in topological confinement, rather than to find the best model for simulating ice between graphene. Even if we were, a careful literature review shows that the uniform confining potential is a reasonably accurate model for simulating water between graphene. And finally, in good faith we are happy to share our preliminary findings from ongoing work that the flat rhombic phase is indeed stable between two layers of graphene

(with real carbon atoms).

We very much appreciate the rest of the comments that improve the flow and figures of the manuscript.

Comments:

The use of the term 'graphene-like nanocavity' (along with other references to graphene) seems somewhat misleading, considering that the hydrophobic confinement in the simulations is created by perfectly rigid, smooth hydrophobic walls (i.e., the potential is 'uniform in the plane of confinement'). While the Morse potential was parameterized based on water-carbon interactions, the specific arrangement of carbon atoms in the hydrophobic wall would likely have a crucial impact on the stability of the ice phases at this level of confinement.

We acknowledge the concerns raised by the reviewer. The perfectly smooth walls serve as a simple model studying the water-graphene interface. However, this model provides a good description of ice phases compared to nanoconfinement simulations that explicitly incorporate carbon atoms. For example, simulations that use an "atomistically flat" confining potential in *Physical Chemistry Chemical Physics*, 21(32), 17640-17654, reproduce solid/square-like phases observed in explicit graphene-water nanoconfinement simulations reported in *Nature* 519, 443–445 and *Sci Rep* 2017, 7, 2646. We believe that considering the large body of careful simulation work employing these models, it is reasonable to refer to our confinement model as "graphene-like." We have made this point clear in the manuscript.

"Such a potential, characterized by perfectly smooth walls, has been widely adopted in previous studies, both in force field [9–11] and first-principles research [12, 14–17, 59]. The uniform confinement potential has demonstrated semi-quantitative accuracy in describing the behavior of water confined within graphene-like cavities [11], as corroborated by a good agreement of stable phases and melting temperatures with respect to confinement simulations that include explicit carbon atoms [48]."

On a related note, I wonder if a perfectly rigid and perfectly smooth wall is required to stabilize this unusual ice phase. I maintain certain skepticism about whether this phase would be stable when a real hydrophobic material (e.g., graphene) is used as confining wall. For instance, the out-of-plane deformations of the confining surfaces have been shown to play an important role in the structure of nanoconfined water (e.g., *ACS Nano* 2018, 12, 1, 448–454; *Sci Rep* 2017, 7, 2646). The authors should address whether the realistic confinement conditions detailed in those, and similar studies would impact their findings.

We appreciate the reviewer's concern. However, we wish to highlight the fundamental importance of our results obtained using an "atomistically flat" confining potential. Such a potential allows an understanding of topological confinement effects on water—a phenomenon that extends beyond the specific context of water-carbon interactions. The simplicity of an atomistically flat potential allows a cleaner interpretation of these effects, which might get obfuscated by the role of interactions in real materials. We have emphasized this point in the main text.

“Furthermore, the uniform confinement model’s atomistically flat nature allows for a clean interpretation of topological confinement effects – a phenomenon that extends beyond the specific context of graphene-based confinement.”

At the same time, we maintain that the "atomistically flat" confining potential is robust in modelling the behaviour of a flexible graphene interface at finite temperatures. To address the reviewer's comments, we have carefully considered the papers cited by the reviewer to check whether the finite-temperature motion of carbon atoms would qualitatively change our results.

Sci Rep 2017, 7, 2646 reports explicit simulations of water encapsulated between graphene sheets, and it stabilises a rhombic phase of ice with a high melting temperature of 480–490 K. For the same (SPC/E) water model, Physical Chemistry Chemical Physics, 21(32), 17640-17654 also report rhombic ice that melts at 577 K. While the agreement isn't quantitative it is remarkable how robust the simple confining potential is in qualitatively describing the phase behaviour of water carbon nanoconfined system w.r.t simulations that include fully flexible graphene sheets beyond room temperatures. We have emphasized this in the main text.

“The uniform confinement potential has demonstrated semi-quantitative accuracy in describing the behavior of water confined within graphene-like cavities [11], as corroborated by a good agreement of stable phases and melting temperatures with respect to confinement simulations that include explicit carbon atoms [48].”

ACS Nano 2018 reports interesting simulation studies of multilayer water graphene interfaces and compares the role of the flexibility of carbon atoms. However, our simulation setup doesn't correspond to this system, which applies a hydrostatic pressure on multiple graphene layers in water. On the other hand, our setup resembles the system in Sci Rep 2017, 7, 2646 with the termination of graphene sheets and vdW attraction, implying a lateral pressure on the system. To clarify the system our setup corresponds to we have explicitly stated the following:

“The nanoconfined system considered in this work is a single layer of water molecules trapped between two parallel sheets – mimicking the experimental setup in Ref. 27.”

And finally we are able to inform the reviewers in good faith that preliminary results from ongoing work indeed suggest that the flat-rhombic phase maintains its stability in the presence of real carbon atoms.

I question whether the lateral pressures simulated (e.g., 2 GPa) align with what could realistically be generated by ‘vdW confinement’. Ultimately, if there is a possibility of observing this phase in laboratory conditions, it is essential we understand the specific conditions under which it could be expected to occur.

We appreciate this concern but there exists a clear consensus that graphene based confinement leads to lateral pressures in the GPa scale. As reported in Nature 519, 443–445, it is possible to estimate in a nice back-of-the-envelope style calculation that shows that this lateral force in graphene nanocapillaries should be on the gigapascal scale. This estimate is independently verified in Sci Rep 2017, 7, 2646 through explicit simulations of water encapsulated between graphene sheets. There also exist experiments suggesting that irradiation of graphene with Ar leads to nanobubbles that can exert lateral vdW pressures in the tens of GPa regime.

Ref. 17 shows that the flat rhombic phase is stable for a fairly wide window of pressure in the 0.5 to 4 GPa range. Furthermore, the authors of Ref. 18 independently predicted a similar structure as being stable in this range of lateral pressures (although they consider a slightly wider nanoconfinement width). Hence, we feel confident that this is indeed the stable phase under lateral pressures that we would expect between two sheets of graphene. We have added the following text to the main text to clarify this point:

“Both simulations [48] and experiments [46] have estimated this vdW pressure to be on the gigapascal scale. Hence, our simulations are run at a lateral pressure of 2.0 GPa across various temperatures within the metastability range of the flat-rhombic phase [16].”

The authors' use of italics seems stylistically inconsistent. At times, italics are used for emphasis, such as in 'just two hydrogen bonds per water molecule,' and at other times, seemingly for qualifiers, like 'covalently bonded.' A more consistent application of italics for specific purposes would enhance the readability.

We thank the reviewer for flagging this. After looking through the manuscript again, we agree that the use of italics was not stylistically consistent, and we have gone ahead and made the use of italics more consistent.

There is a typo in the caption of Table I, it mentions '... the lattice energy, *E_{chain}*,' which should correctly be '*E_{lattice}*.'

We thank the reviewer for their attention to detail. This has now been corrected.

“The energy values include the total and vdW contributions to the lattice energy *E_{lattice}*, the cohesive energy of a quasi-one-dimensional chain of water molecules *E_{chain}*, and the remaining stabilization energy from interactions between chains of water molecules *E_{stack}*.”

The discussion involving percentages in the right column of Page 3 is somewhat confusing, primarily because the relationship between the different 'types' of binding energies selected by the authors isn't immediately clear. Including at least the equation $E_{stack} = E_{lattice} - E_{chain}$ from the methods into the results section—or possibly the entire relevant paragraph—would significantly improve the paper's flow.

We thank the reviewer for the suggestion. We have included the $E_{\text{stack}} = E_{\text{lattice}} - E_{\text{chain}}$ equation along with our qualitative description of the different stabilization energies, although we keep the formal definitions in the Methods section.

“... This means that the stacking energy is the stabilization energy of the lattice that is not captured by the chains alone: $E_{\text{stack}} = E_{\text{lattice}} - E_{\text{chain}}$.”

I believe the characterization of the hexagonal ice phase as having only a single chain motif is inaccurate. Similar to graphene, there should be 'zigzag' and 'armchair' directions, which would imply the existence of two distinct chain types.

We agree with the reviewer. We have included results for the armchair chain too. We have also appropriately updated Fig. 1 with the average binding energies and added the following text to the main text:

“In the hexagonal phase, we consider two distinct chain types in the ‘zigzag’ and ‘armchair’ directions and report the average binding energies.”

The legend in Figure 1b might be misleading or possibly incorrect. It's unclear whether the fully colored section of the bars represents the 'Total' or 'Total – vdW' energy. I assume it's the latter. If so, please consider relabeling for clarity.

The filled bars represent the total energy and the hashed bars represent the vdW energy only (so that their percentual contribution can be seen by eye). To avoid confusion, these have now been separated into two independent bars.

A clear and precise definition of the angles theta and phi is missing in the text. I find the graphical illustration in Figure 2a confusing, and the 'intuitive' explanation of their meaning in the caption of Figure 2 unhelpful. Please, specify the vector and the plane against which these angles are defined.

We appreciate this feedback. Since the definitions of the theta and phi angles are extremely important, we have gone ahead and added an additional illustrative figure to the main text (Figure 2) to aid in clarity towards how these angles are defined. We have also moved the formal definitions of these angles from the appendix into the main text. We hope that these changes have made the definition of these angles more clear.

I find Figure 2a quite confusing. The stability of a given configuration is determined within a specific region of the (theta, phi) space, yet only one angle is depicted in each panel. It would also be helpful to clarify the precise criteria used for determining stability.

We have gone ahead and separated out some of the information from the original Figure 2 from the previous manuscript and put it into the new added figure in the main text. We believe that the new version (with Figure 2 and Figure 3 separated) helps clarify our message.

I understand that the angle ϕ is periodic, with a period of $\pi/2$. Could you explain why the plots extend from $-\pi/2$ to $\pi/2$ (instead of just 0 to $\pi/2$)?

We believe this confusion arose from the lack of clear definitions of θ and ϕ angles. Hopefully the new figure and the added descriptions in the main text have clarified this point. The ϕ angle is indeed periodic with a period of π , as indicated by the plot. We hope that the following addition to the main text clarifies this point:

“The $\phi = \pi/2 - \arccos(\hat{\mu} \cdot \hat{z})$ angle captures the alignment between the molecular dipole vector μ and the z -axis, with hats above vectors indicating normalized unit vectors. A water molecule with $\phi = \pm\pi/2$ will have its dipole vector aligned with the $\pm z$ -axis. We also wrap around the ϕ angles computed such that ϕ always lies in the range $[-\pi/2, +\pi/2]$.”

The labels in Fig 2b for the states illustrated in Fig 2a obscure the data. While I question the necessity of highlighting these states, if the authors choose to include them, an alternative presentation should be sought to avoid hiding the data.

We agree with this feedback- the labels obscure the free energy minima themselves. The updated separated figures have accordingly been updated to prevent this.

To enhance the clarity and flow of the paper, it might be beneficial to include a separate panel for the number of putative hydrogen bonds shown in Figure 2b, right column.

With the creation of a new figure for the left panel of the original Figure 2, we hope that the updated Figure 3 in the new manuscript is more clear. We have also included panel labels which help the discussion of the figure.

There is a crucial discussion about the connection between regions of low probability and a higher number of putative hydrogen bonds. However, the data in Figure 2 presents $\log(p)$ and the number of hydrogen bonds independently, as functions of ϕ and θ . The manuscript would greatly benefit from a density plot illustrating the correlation between $\log(p)$ and the number of hydrogen bonds. Without this, assessing the validity of the authors' observations becomes challenging.

We thank the reviewer for this suggestion. The main idea of our claims around the $\log(p)$ and number of hydrogen bonds plots concerns the location of the probability maxima and how they relate to the maxima and minima in the number of hydrogen bonds plot. These are really rather small portions of the entire 2D histogram of ϕ and θ angles. The requested correlation plots would be almost entirely full of data points that correspond to points outside of these probability maxima, which we believe might obscure our primary message. Our specific comments are towards the fact that the probability maxima do not align with the maxima in the number of hydrogen bonds plots and that the stark difference in these profiles is both unusual and unexpected.

The authors state, 'the peculiar hydrogen bonding behavior [...] emphasizes the importance of incorporating dynamical information into the hydrogen bond definition.' However, from my understanding, the authors do not undertake such an approach but rather adhere to the geometric 'canonical' definition used by Chandler. Could you clarify the rationale behind emphasizing the need for dynamical information in the hydrogen bond definition?

We thank the reviewer for this comment. The issue with the geometric definition is that it can count hydrogen bonds even when the lifetimes are shorter than typical intermolecular oscillations — see *Angewandte Chemie International Edition*, 59(42), 18578–1858. To clarify this issue, we have added a figure in the Supplementary Information where we compute the number of “dynamical” hydrogen bonds as a function of temperature.

We have also removed the definition of hydrogen bonds from the methods section. And instead we explicitly state whether we used the geometric or dynamical definitions throughout the text.

“In Fig. 4(a) of the main text, we showed that employing such a geometric hydrogen bond definition resulted in the number of hydrogen bonds seeming to increase as temperature is raised. This contradicts the expected behavior from a standard energy-entropy trade-off perspective. We suggested that this was because this geometric definition still included fleeting hydrogen bonds that only lasted for times shorter than intermolecular timescales. Employing the definition by Schienbein and Marx [39], we also computed the number of dynamical hydrogen bonds as a function of temperature. Here, a dynamical hydrogen bond is one that satisfies the above geometric criteria for at least 0.2 ps, the typical timescale for intermolecular oscillations for monolayer ice as well as supercritical and room temperature water [39]. The resulting dynamical hydrogen bond count exhibits the expected decreasing trend, as shown in Supplementary Figure 3.”

The colors used in Figure 3b are not easy to distinguish, and the symbols are the same for all the series.

The authors are encouraged to show the error bars in the calculation of the h-bond lifetime presented in Fig 3c.

We have updated the colors in the new Figure 4(b) to be easier to distinguish. However, we prefer to keep the symbols the same for all the series. We chose a diamond shape because it describes the flat-rhombic phase (to distinguish it from the other phases considered in the SM), and we would prefer that there is only one distinguishing feature between the different data point types: in this case, it is the color.

We thank the reviewer for the suggestion, and we agree that the error bars add valuable information. We have provided updated plots in Fig. 4(c) that have the error bars included.

A considerable portion of the text is dedicated to discussing the dielectric constant of the ice phase, yet the relevant figure is only included in the SI. It might be more appropriate to feature this figure in the main text, or alternatively, consolidate the discussion about it.

We thank the reviewer for this suggestion. We have incorporated the section into the main text.

“A consequence of the quasi-one-dimensional hydrogenbonded structure is that it may possess ferroelectric behavior due to its net dipole moment along the direction of the hydrogen-bonded chains. Ferroelectricity in ice has been conjectured previously in force field simulations [41], first-principles calculations of nanoconfined water [42], and experiments on confined [28] and supported [43] films. To investigate the dielectric behavior of flat-rhombic ice, we model the temperature dependence of its dielectric response in Fig. 5. We analyzed the in-plane ϵ_{\parallel} and out-of-plane ϵ_{\perp} dielectric constants based on the variance of a linear polarization model: the TIP4P point charge model [44]. We report the classical dielectric response, as quantum nuclear effects are expected to only make a small quantitative difference [44].”

“We observe low and near-constant in-plane and out-of-plane dielectric constants for the flat-rhombic phase, in agreement with the experiments on nanoconfined water from Ref. 45. The dielectric nature of flat-rhombic ice remains the same from 0 K up to its phase transition into the paraelectric hexatic (disordered) water phase [16]. The temperature independence of the flat-rhombic phase’s dielectric behaviors reflects the resilience of the quasi-one-dimensional structure to thermal and quantum nuclear fluctuations. The singularity observed in Fig. 5 indicates the temperature at which the phase transition to the hexatic phase occurs. The hexatic phase exhibits an increased in-plane dielectric constant but a similarly small out-of-plane dielectric constant.”

The authors transition from discussing findings in a system of 576 molecules to one with 144 molecules, justifying this shift to the observation of concerted dynamics in the smaller systems. If such dynamics are not observed in the larger system, should there not be concerns regarding artifacts originated from finite size effects? This aspect appears troubling.

Our results for 144 molecules remain significant as we are not aiming to understand coherent proton disorder in a thermodynamically large system. Instead, we are focusing on a specific situation when our simulation cells correspond to sizes of nanocavities that are experimentally accessible. In particular, the length scale considered in our simulations corresponds to nanopockets of a few nanometers in dimensions when water is encapsulated between graphene sheets. We have made this point more evident in the main text:

“However, crystallites of nanoconfined water molecules containing around 100 water molecules are experimentally accessible, so such concerted motion could be observed in experiments as small ice crystallites forming between graphene sheets [46] or high-pressure graphene nanobubbles formed by irradiation [47]. Previous work has suggested that the simulation setup we employ is still reasonable for studying such encapsulated systems, even without explicit confining atoms [46, 48].”

The presentation of the data in Figure 4b makes the behavior it intends to reveal unclear. The authors are encouraged to consider a different style of plotting, such as stacked series, to ensure the time evolution is clearly discernible across the three temperatures.

We thank the reviewer for this suggestion. The updated version in Figure 6(b) contains separate panels for the time series of the order parameters at different temperatures so that the different behaviours are clearly discernible.

The findings depicted in Figure 4 appear very intuitive but their connection to coherent proton dynamics is not immediately clear to me. Wouldn't an analysis of dynamical correlations would be necessary to assess this connection?

As part of our response to reviewer #2 we have now included a video. We think this addresses this comment too. If the protons become disordered, then the sigma order parameter will be near 0 instantaneously. We have also added the following sentences to the main text so that this point is clearer in the text.

“If the motion of the protons is not correlated, then the values of the ϕ_{ij} values will be uncorrelated with the row index i . This will cause the σ value to take on intermediate values near 0, which is indicative of a loss of proton order.”

The authors refer to the 'observed length scale of concerted motion' in the last paragraph of the Results section, but I find this confusing as there is no prior discussion or presentation of results related to length scales in the manuscript.

We agree that it may have been confusing. We have rewritten this sentence in the context of number of atoms rather than length scales.

“However, crystallites of nanoconfined water molecules containing around 100 water molecules are experimentally accessible, so such concerted motion could be observed in experiments as small ice crystallites forming between graphene sheets [46] or high-pressure graphene nanobubbles formed by irradiation [47]. Previous work has suggested that the simulation setup we employ is still reasonable for

studying such encapsulated systems, even without explicit confining atoms [46, 48].”

Reviewer #1 (Remarks to the Author):

Publish after minor revision.

Comments: I appreciate the authors' efforts in addressing most of the questions I raised. I am curious whether it would be possible to conduct an MD simulation to confirm that ZZ-qBI is unstable in the absence of van der Waals forces. Such a study could further solidify the conclusions regarding the role of van der Waals forces.

Reviewer #3 (Remarks to the Author):

See attachment

Reviewer #3 Attachment on the following page

The authors have made efforts to address the issues raised. A few more issues need to be clarified before I can recommend publication of the manuscript in Nature Communications.

1. The authors used classical TIP4P water model to study the dielectric response of the flat-rhombic phase. The rationale of the specific water model used should be provided as the formation of flat-rhombic phase is model-dependent and cannot be obtained using TIP4P water model. The authors should include an analysis of low dielectric response in other phases as well to show its connections to van der Waals interactions.

2. The statement regarding long-range proton dynamics has been challenged by other referees, in that concerted proton motion exists only in small systems and dissipates in larger systems, or maybe influenced by periodic boundary conditions. So, the reported phenomenon may be due to specific MD settings. If the authors wanted to investigate the concerted proton motion only in small systems, then the finite-size ice island is more appropriate to be studied. Otherwise, the edge effect should be discussed as it may play important role in their relatively small system.

3. I suggest the authors to perform an analysis of rotational motion as the function of the temperature, such as auto-correlation functions, to demonstrate whether there is an abrupt change or a continuous transition from low temperature to the melting point. Here the melting point is 380 K. So the nearly same distribution functions at 380 K and 280 K could only indicate hysteresis in an over-heated state.

4. Regarding the additional 0.2 hydrogen bonds mentioned, are these formed due to hydrogen bonding between two water chains? The authors should comment on the structural origin of these additional hydrogen bonds in the 2D ice phases. A similar analysis should be also made from the DFT-based PIMD simulations.

5. The comparison of energy and force alone may not be sufficient to guarantee the physically meaningful dynamics obtained from the MLP-based MD simulations. Providing AIMD trajectories of the concerted proton motion in both small and larger systems would ensure the validity of the observed phenomenon.

6. The authors acknowledge that the "flat-rhombic" ice formed in the 5-Å confinement and the ZZMI formed in the 6-Å confinement correspond to the same phase. The authors need to fix the typo that the confinement width reported for the ZZMI phase in previous publications (*Nature Physics* v20, 456, 2024) was 6.5 Å. It should be the narrower pore of 6 Å. In the latter, the ZZMI phase seems also very flat. There was another typo "zigzagging" which should correctly be 'zigzag'.

Reviewer #4 (Remarks to the Author):

The authors have thoroughly addressed my concerns, and I believe the manuscript is now ready for publication. I would like to reiterate my commendation for the high technical quality of their work. This manuscript represents a significant contribution to the study of exotic phases under nanoconfinement.

Reviewer #1:

I appreciate the authors' efforts in addressing most of the questions I raised. I am curious whether it would be possible to conduct an MD simulation to confirm that ZZ-qBI is unstable in the absence of van der Waals forces. Such a study could further solidify the conclusions regarding the role of van der Waals forces.

We thank the reviewer for their comment. Our new calculations for the ZZ-qBI phase do indeed suggest that this phase would be *thermodynamically* unstable in an MD simulation run without van der Waals interactions. However, it is still possible that the phase may be *metastable*, so running such a simulation may not directly address this point. This exercise, whilst interesting in principle, may not be informative about the thermodynamic stability beyond our current analysis.

Furthermore, running such a simulation in the presence or absence of van der Waals forces is beyond the capabilities of our currently trained MLP. Our model is trained directly on dispersion inclusive DFT so it isn't possible to switch off van der Waals. In addition, our model isn't trained for pressures exceeding 8 GPa. We do agree, however, future work should investigate the role of vdW interactions to understand thermodynamic and dynamical stability of high pressure confined ice phases (beyond 8 GPa). To make these points clear, we have added the following to the main text:

As our machine learning potential isn't trained for pressures beyond 8 GPa or just revPBE0 level, we are unable to comment on the dynamical stability or metastability of the ZZ-qBI phase in the absence of the vdW interactions.

Reviewer #3:

The authors have made efforts to address the issues raised. A few more issues need to be clarified before I can recommend publication of the manuscript in Nature Communications.

1. The authors used classical TIP4P water model to study the dielectric response of the flat-rhombic phase. The rationale of the specific water model used should be provided as the formation of flat-rhombic phase is model-dependent and cannot be obtained using TIP4P water model. The authors should include an analysis of low dielectric response in other phases as well to show its connections to van der Waals interactions.

We agree with the reviewer that it makes sense to justify the choice of the dipole moment

surface.

We used point charges from the TIP4P model to determine the system's polarization for reasons of computational efficiency and its semi-quantitative accuracy.

The MLP used in our work only gives access to the potential energy surface. All other electronic properties, including the system's polarization, require computationally demanding *ab initio* methods. The dielectric response of a system requires long simulations, e.g. a few nanoseconds for bulk water, implying an exceedingly large number of *ab initio* calculations for the temperature dependent dielectric response profile.

TIP4P point charges on the other hand give a good qualitative and semi-quantitative description of the dielectric response of water. The TIP4P dielectric constant of bulk water at room temperature is around 60 which is around 23 % lower than the experimental value of 78 [1]. This discrepancy arises from the underprediction of the molecular dipole moment in liquid water at 300 K and is due to the lack of electronic polarization. For instance, the molecular dipole at TIP4P level is 2.348 D which is around 12.8 % lower than that estimated at MP2 or MP4 level [2]. Since the dielectric response depends on the square of the dipole, the lack of electronic polarization should result in nearly a 24 % lower dielectric response – in excellent agreement with the 23 % lower prediction compared to experiments. Accounting for electronic polarization, by multiplying by a factor of ~ 1.25 [3], do not significantly change our results and conclusions.

To address the reviewer's concern, we have rescaled our dielectric responses by a factor of 1.25, as is typically done, to obtain an approximate *ab initio* dielectric response. We have included the following sentence in the main text

Since our MLP only predicts the potential energy surfaces, a first principles investigation of the dielectric response would be computationally demanding due to numerous single point calculations of the electronic polarization [45]. Therefore, for a semi-quantitative understanding, we use a simple linear polarization model based on TIP4P charges [46].

We select the TIP4P water model due to its low computational cost and a semi-quantitative description of the dielectric response of bulk [46] and confined [47] water. The TIP4P model underpredicts the polarization of aqueous systems as it doesn't incorporate the electronic polarization of the water molecules. For instance, TIP4P predicts a molecular dipole moment of 2.348 D which is 12.8% lower compared to first principles estimates [48], and effectively leads to a $\sim 25\%$ lower dielectric response. To incorporate the lack of the electronic polarization in the TIP4P model, we rescale the calculated dielectric response by a factor of 1.25, as has been done previously [47].

We sincerely believe that studying other phases, although interesting, is beyond the scope of this study which primarily focuses on the pressure regime accessible in experiments. However, we do understand where the reviewer's concerns arise from, hence we have included a statement that the low dielectric response of this phase is not a direct outcome of the vdW interactions but an indirect one as vdW interactions are responsible for the thermodynamic stability of this phase.

The low dielectric response of this phase is only an indirect outcome of vdW interactions, as they dictate the thermodynamic stability of the flat rhombic phase.

[1] Scott Habershon, Thomas E. Markland, and David E. Manolopoulos, "Competing quantum effects in the dynamics of a flexible water model," *The Journal of Chemical Physics* 131, 024501 (2009)

[2] Anna V. Gubskaya and Peter G. Kusalik, "The total molecular dipole moment for liquid water," *The Journal of Chemical Physics* 117, 5290–5302 (2002).

[3] T. Dufils, C. Schran, J. Chen, A. K. Geim, L. Fumagalli, and A. Michaelides, "Origin of dielectric polarization suppression in confined water from first principles," *Chemical Science* 15, 516–527 (2024)

2. The statement regarding long-range proton dynamics has been challenged by other referees, in that concerted proton motion exists only in small systems and dissipates in larger systems, or maybe influenced by periodic boundary conditions. So, the reported phenomenon may be due to specific MD settings. If the authors wanted to investigate the concerted proton motion only in small systems, then the finite-size ice island is more appropriate to be studied. Otherwise, the

edge effect should be discussed as it may play important role in their relatively small system.

We thank the reviewer for their comment. We agree that for this kind of small water pocket confined between two graphene sheets, an exact characterization of the experimental dynamical properties would require a careful treatment of the water-carbon interactions near the edges of the pocket. Although, we have already stated in the paper and in our previous response that the simple uniform confining potential is a good model for studying water encapsulation, we agree that in small water pockets one can have quantitative deviations due to the edges of the pockets, beyond the lateral pressure. To clarify this important point, we have included the following sentences:

A careful treatment of such encapsulated systems would require a detailed analysis of the water-carbon interactions at the edge of the confined water pockets. In this work, we ignore these edge effects and instead focus on characterizing the behavior of nanoconfined water under idealized atomistically flat nanoconfinement. A careful consideration of the edge effect would be an interesting and relevant topic for future work.

3. I suggest the authors to perform an analysis of rotational motion as the function of the temperature, such as auto-correlation functions, to demonstrate whether there is an abrupt change or a continuous transition from low temperature to the melting point. Here the melting point is 380 K. So the nearly same distribution functions at 380 K and 280 K could only indicate hysteresis in an over-heated state.

We thank the reviewer for this suggestion. The rotational legendre autocorrelation functions of O–H bonds and the associated relaxation times provide a very clear picture of rotational dynamics. We have estimated the $n=1$ rotational relaxation autocorrelation functions and included a plot and discussion in the SI (see below). We observe an Arrhenius-like trend for the temperatures between 280 and 360 K i.e. the conditions where we observe the concerted rotational motion of water molecules.

For dynamical insights into the rotational motion of water molecules, we estimate the rotational Legendre autocorrelation functions of the O–H bond with $n = 1$, following the procedure described in Ref. [80]. This autocorrelation function will decay to 0 as the initial rotational configuration of the water molecule relaxes to a random orientation. For each displayed temperature in Supplementary Figure 4(a), we estimate the autocorrelation function over a 25 picosecond window, averaged over 5 trajectories. The definite integral of this function, from 0 to ∞ , gives the rotational relaxation time.

Supplementary Figure 4(a) clearly shows two different types of rotational motion in the flat-rhombic phase. As we mentioned in the discussion surrounding Fig. 3 of the main text, water molecules in low-temperature conditions remain localized in their molecular orientations up to the simulation time considered in this work. At low temperatures such as 160 and 220 K, this behaviour leads to a sharp initial decay in the rotational Legendre autocorrelation plots in Supplementary Figure 4(a). This sharp decay is due to thermal fluctuations within the (θ, ϕ) free energy minimum. As a result, these autocorrelation functions exhibit a nearly infinite rotational relaxation time, since the water molecules are stuck in the free energy minimum that they begin in. The diverging relaxation time is an artefact of the absence of molecular rotation sampling within the timescales of our simulations. At intermediate temperatures such as 280 K and 360 K, we observe finite rotational relaxation times, consistent with the (θ, ϕ) exploration of water

molecules in Fig. 3 of the main text. We also note a good fit of the rotational relaxation times between 280 K and 360 K to an Arrhenius equation, suggesting a common activation barrier dictating the rotational motion. We do not include the temperatures 160 and 220 K in the fit, as at these temperatures we do not sample the relevant relaxation event, implying an ∞ relaxation time. We do not account for quantum nuclear motion as they are expected to only make a quantitative difference to this trend [46, 81]. Specifically in the context of the flat-rhombic phase, the correspondence between Fig. 3 of the main text and Supplementary Figure 4, and the quantitative increase in the exploration of (θ, ϕ) orientations due to quantum nuclear effects (see Supplementary Figure 7) implies a systematic increase the relaxation time.

These results further bolster our point that the rotational behavior at 360 K in Figures 3 and 6 of the main text arises from thermal activation in the metastable phase of the solid beyond the melting temperature. We further note that this observation is in line with the reviewer's assertion as "a metastable solid phase beyond the melting temperature", as currently discussed in the main text, is tantamount to "hysteresis in an overheated phase".

4. Regarding the additional 0.2 hydrogen bonds mentioned, are these formed due to hydrogen bonding between two water chains? The authors should comment on the structural origin of these additional hydrogen bonds in the 2D ice phases. A similar analysis should be also made from the DFT-based PIMD simulations.

The additional 0.2 'putative' hydrogen bonds indeed arise from hydrogen bonding between water chains in the flat-rhombic phase. This origin of these hydrogen bonds is the same as the emergence of 'putative' hydrogen bonds at finite temperature. Although these points are already discussed in the manuscript, we appreciate that these conclusions may not be immediately clear. Hence, to make the discussion on quantum nuclear motion clearer, we include the following:

In these configurations, hydrogen bonds form between different chains in the flat-rhombic crystal structure, as they do classically at high temperatures. The increased disorder due to quantum nuclear effects is not surprising, as the effect of quantum nuclear motion on properties of water has often been mapped to a temperature increase [40]... Using this notation, we see that this increase in the number of putative hydrogen bonds can be attributed to the increasing prevalence of instantaneous 2D1A motifs, i.e. instantaneous hydrogen bonds between chains.

5. The comparison of energy and force alone may not be sufficient to guarantee the physically meaningful dynamics obtained from the MLP-based MD simulations. Providing AIMD trajectories of the concerted proton motion in both small and larger systems would ensure the validity of the observed phenomenon.

We appreciate the reviewer's concern and we agree that in principle energy and force errors do not completely dictate errors in dynamics or sampling.

Aware of this issue, already in our previous work (Ref. [2]), we reported “out of sample” errors by performing MLP simulations at every temperature and pressure, sampling configurations and reporting the error against DFT as a function of temperature and pressure (See Extended Data Fig. 1 panel c of the paper cited above). We reported very small errors in forces, implying a strong degree of confidence in real time or thermostatted dynamics. Indeed, the prediction of our MLP simulations were confirmed against *ab initio* simulations.

In this work, we extend the training set with additional configurations and external conditions and perform the same “out of sample” validation as in Ref. [2]. The errors reported in Appendix A 2 reflect a high degree of confidence in the system’s dynamics. Noting that our errors are consistent with our previous work and work done by others (Refs. [1-3]), we are confident in the predictive ability of our MLP. This point is addressed in the following paragraph from our last update to the manuscript:

To assess the neural network’s ability to generalize to unseen configurations, we compared the energies and forces produced by the MLP to those produced by the reference revPBE0-D3 functional on new structures that were not seen during training time. These structures were generated by running 2 nanosecond N Pxy T simulations at 2.0 GPa starting from each of the solid phases identified in Ref. 16. Each solid phase was simulated at 100 K, 300 K, and 600 K. For some of these simulations, the solid phase was metastable, and so the trajectory remained in the solid phase. For other simulations, the solid phase was unstable, and the water molecules in the simulation entered a disordered phase or glassy phase (depending on the conditions). Hence, these trajectories contained a diverse set of ordered and disordered configurations that span a broad range of local molecular configurations. We computed the errors between the MLP’s predictions and the reference revPBE0-D3 calculations over all of the configurations generated in each of these trajectories. The resulting mean absolute errors (MAE) for the energies and root-mean-squared errors (RMSE) for average atomic forces are shown in Table II. These errors are similar in magnitude to the errors of MLPs employed in other studies [16, 71, 72].

Moreover, over the past few years, we have demonstrated the accuracy of MLPs for estimating real time dynamics properties of water (see Refs. [4, 5, 6]) with quantum nuclear motion and compared directly against DFT references (see Ref. [7]). Despite the lack of stringent “out of sample” validations, we have always observed an excellent description of aqueous dynamics with the framework of Behler-Parrinello neural networks. To highlight this in the manuscript we include the following statement in the Supplementary Information:

We also note that this model [41] (and its predecessors based on the same machine learning architecture [69, 70]) have been validated against experimental IR, Raman and sum frequency generation spectra, demonstrating an excellent description of real-time dynamics of aqueous systems.

And finally we also note the continued agreement between our results and the results of independent work using independently-trained MLPs [1] – which have been directly benchmarked against *ab initio* simulations – suggests that the training and predictive ability of aqueous MLPs is robust with respect to thermodynamic conditions. To highlight this in the manuscript we include the following statement in the supporting information

These errors are similar in magnitude to the errors of MLPs employed in other studies [16, 17], which have been validated against reference DFT simulations.

[1] Bo Lin, Jian Jiang, Xiao Cheng Zeng, and Lei Li, “Temperature-pressure phase diagram of confined monolayer water/ice at first-principles accuracy with a machine-learning force field,” *Nature Communications* 14, 4110 (2023).

[2] Venkat Kapil, Christoph Schran, Andrea Zen, Ji Chen, Chris J. Pickard, and Angelos Michaelides, “The first-principles phase diagram of monolayer nanoconfined water,” *Nature* 609, 512–516 (2022).

[3] Marco Eckhoff and Jorg Behler, “Insights into lithium manganese oxide–water interfaces using machine learning potentials,” *The Journal of Chemical Physics* 155, 244703 (2021).

[4] Venkat Kapil, David M. Wilkins, Jingtang Lan, and Michele Ceriotti, “Inexpensive modeling of quantum dynamics using path integral generalized Langevin equation thermostats,” *Journal of Chemical Physics* 152, 124104 (2020).

[5] Sam Shepherd, Jingtang Lan, David M. Wilkins, and Venkat Kapil, “Efficient Quantum Vibrational Spectroscopy of Water with High-Order Path Integrals: From Bulk to Interfaces,” *Journal of Physical Chemistry Letters* 12, 9108-9114 (2021).

[6] Venkat Kapil, David P. Kovács, Gábor Csányi, and Angelos Michaelides, “First-principles spectroscopy of aqueous interfaces using machine-learned electronic and quantum nuclear effects,” *Faraday Discussions* 246, 313-339 (2023).

[7] Ondrej Marsalek and Thomas E. Markland, “Quantum Dynamics and Spectroscopy of *Ab Initio* Liquid Water: The Interplay of Nuclear and Electronic Quantum Effects,” *Journal of Physical Chemistry Letters* 8, 1545–1551 (2017).

6. The authors acknowledge that the "flat-rhombic" ice formed in the 5-Å confinement and the ZZMI formed in the 6-Å confinement correspond to the same phase. The authors need to fix the typo that the confinement width reported for the ZZMI phase in previous publications (*Nature Physics* v20, 456, 2024) was 6.5 Å. It should be the narrower pore of 6 Å. In the latter, the ZZMI

phase seems also very flat. There was another typo “zigzagging” which should correctly be 'zigzag'.

We thank the reviewer for pointing out these mistakes - we have corrected them in the updated manuscript.

Reviewer #4:

The authors have thoroughly addressed my concerns, and I believe the manuscript is now ready for publication. I would like to reiterate my commendation for the high technical quality of their work. This manuscript represents a significant contribution to the study of exotic phases under nanoconfinement.

We thank the reviewer for their appreciation.

Reviewer #3 (Remarks to the Author):

The authors have mostly addressed reviewers' comments.

I'd like to recommend publication of the paper in Nature Communications after the authors make the final minor change on page 4:

When the authors discussed the zigzag monolayer ice (ZZMI) on page 4, only Ref. 17 (published in 2023) was cited. Actually, the ZZMI was first reported in Ref. 15 (published in 2021). So, for the respect of historical development, the authors should also cite Ref. 15 when discussing ZZMI on page 4.

Below is our response to the final comment by Reviewer #3 (in red), followed by an example of how we have updated the text to reflect the reviewer's comment on references (in blue).

Reviewer #3 (Remarks to the Author):

The authors have mostly addressed reviewers' comments. I'd like to recommend publication of the paper in Nature Communications after the authors make the final minor change on page 4:

When the authors discussed the zigzag monolayer ice (ZZMI) on page 4, only Ref. 17 (published in 2023) was cited. Actually, the ZZMI was first reported in Ref. 15 (published in 2021). So, for the respect of historical development, the authors should also cite Ref. 15 when discussing ZZMI on page 4.

We thank the reviewer for pointing this out- we have carefully gone through our citations to Ref. 15 and Ref. 17 to ensure that they are consistent with this, and we now cite both when we discuss the ZZMI phase. An example is below (although we have made the citation updates everywhere else as well):

"Refs. 15 and 17 explored even higher lateral pressures for a confinement width of 6 Å. These studies found that the stable phase beyond 0.5 GPa is a zigzag monolayer ice (ZZMI) phase, which is topologically equivalent to the flat-rhombic phase, except that the larger confinement width leads to a small buckling of the oxygen atoms."